# Hidden information on protein function in censuses of proteome foldedness

Dezerae Cox [1], Ching-Seng Ang[2], Nadinath B. Nillegoda [3], Gavin E. Reid[1,4] & Danny M. Hatters [1 ✉]

Methods that assay protein foldedness with proteomics have generated censuses of apparent protein folding stabilities in biological milieu. However, different censuses poorly correlate with each other. Here, we show that the reason for this is that methods targeting foldedness through monitoring amino acid sidechain reactivity also detect changes in conformation and ligand binding, which can be a substantial fraction of the data. We show that the reactivity of only one quarter of cysteine or methionine sidechains in proteins in a urea denaturation curve of mammalian cell lysate can be confidently explained by a two-state unfolding isotherm. Contrary to that expected from unfolding, up to one third of the cysteines decreased reactivity. These cysteines were enriched in proteins with functions relating to unfolded protein stress. One protein, chaperone HSPA8, displayed changes arising from ligand and cofactor binding. Unmasking this hidden information using the approaches outlined here should improve efforts to understand both folding and the remodeling of protein function directly in complex biological settings.

[1] Department of Biochemistry and Pharmacology, Bio21 Molecular Science and Biotechnology Institute, The University of Melbourne, Parkville, VIC 3010, Australia. [2] Melbourne Mass Spectrometry and Proteomics Facility, Bio21 Molecular Science and Biotechnology Institute, The University of Melbourne, Parkville, VIC 3010, Australia. [3] Australian Regenerative Medicine Institute, Monash University, Clayton, VIC 3800, Australia. [4] School of Chemistry, The University of Melbourne, Parkville, VIC 3010, Australia. ✉email: dhatters@unimelb.edu.au

The maturation of an active protein is typically reliant upon the nascent polypeptide folding into a complex topology. Folding involves a thermodynamic component, which describes the free energy difference between the folded state and unfolded state at equilibrium (folding stability, $\Delta G$). Folding also involves other sequential processing steps, such as post-translational modifications and transport. Because folding is fraught with potential mishaps including misfolding and aggregation, in cells a proteostasis network oversees all steps related to synthesis, folding and degradation[1]. In mammalian cells this network consists of several hundred proteins, including molecular chaperone families (e.g., heat shock protein families 40, 70 and 90), the ubiquitin-proteosome system, autophagy and stress response systems[2].

Proteostasis imbalance is implicated in diseases involving inappropriate protein aggregation, including neurodegeneration[3]. As such, there has been extensive interest in determining how protein foldedness varies for proteomes inside cells in healthy and disease contexts[1,4,5]. A canonical approach for measuring protein folding stability of a purified protein involves measuring the abundance of folded and unfolded states in different concentrations of chaotropes, such as urea or guanidine hydrochloride, or exposure to increasing temperature. This approach yields measures of $\Delta G$ or other correlates of $\Delta G$ such as chemical denaturation midpoint ($C_m$) or thermal melting midpoint ($T_m$) values that are informative in the case where the protein folds by a two-state equilibrium mechanism. Recent advances have allowed this canonical strategy to measure protein folding stabilities in biological extracts, thereby enabling the en masse determination of folding stabilities of proteins[6–13]. Measurement strategies have targeted differences between the folded and unfolded states, such as aggregation propensity, sensitivity of solvent exposed amino acid sidechains to reactive chemicals, which we hereon refer to as residue labeling, and protease cleavage susceptibility[14].

While the main thrust of many proteomics studies employing these approaches has been to test the effect of ligand binding on protein stability rather than define the $\Delta G$ per se, it remains generally unclear how well the reported measures of stability actually relate to the thermodynamic stability of the protein. In biological settings measures of protein folding stability are affected by conformational change, ligand binding and protein network organization. For example, $T_m$ values act as proxies for enzyme activity, DNA binding, and complex formation as well as posttranslational modifications during the cell cycle[13]. In other work, ATP was shown to increase the resistance of thermal denaturation of positively charged, intrinsically disordered proteins, and this resistance depends on their localization to different membrane-less organelles[15]. Limited proteolysis-based mapping of protein structure has also yielded information on metabolite-induced conformational changes of proteins[16]. Because measures of folding stability can act as proxies for other distinct features of proteins, a strategy to more completely decipher whether the measured values of stability relate to the actual conformational stability versus a conformational change or ligand binding event would greatly aid the ability to interpret complex in situ denaturation datasets.

Here, we investigate this problem at two levels. First, we examined how well published datasets correlated with each other to establish whether the folding stabilities are indicative of genuine $\Delta G$ values. Second, we investigated how well the measured values of $T_m$, $C_m$ and $\Delta G$ of individual cysteine and methionine residues within proteins correlate to the actual stability using residue labeling approaches. Our work provides a platform to quantitate the extent through which these values report on additional complexity in the data. We find surprising information related to chaperone activity on a urea denaturation curve of mammalian lysate. The approaches we describe provide a pathway to unmask hidden information pertaining to protein function in complex biological milieu.

## Results

**Methodological differences yield poorly correlated measures of thermodynamic proteome stability.** First, we examined 20 datasets from 12 studies that reported high-quality protein folding stability data, which used limited proteolysis, residue labeling and thermal profiling methods to assay foldedness (complete reference details are provided in Supplementary Table 1). These studies reported $T_m$ or $C_m$ values which, for two-state folding models indicate the conditions of equal populations of unfolded and folded protein (i.e., where $\Delta G$ equals zero). We hypothesized that if these values authentically report on two-state protein folding stability the different datasets should correlate with one another. To test this hypothesis, we first scaled each $T_m$ or $C_m$ dataset to range between 0 and 1 to account for the inherently different magnitudes of $T_m$ and $C_m$ values, and then performed pairwise comparisons between datasets. Linear regressions fitted to each comparison revealed a strong positive relationship in stabilities when comparing datasets derived from the same methodology, particularly among thermal profiling data (Fig. 1; lower diagonal datasets 7–20). This conclusion was not dependent on the species from which the protein stability was measured, suggesting that closely related proteins from different species behave similarly in terms of the $T_m$ and $C_m$ values. However, comparisons between the datasets derived from different methodologies, such as thermal profiling versus residue labeling, revealed poor correlations at best and none at worst. This lack of correlation was supported by Spearman's correlation coefficients calculated for each comparison (Fig. 1; upper diagonal), whereby significant positive correlations were primarily observed between datasets derived from the same methodology. One notable exception was residue labeling dataset 4[17], which was moderately correlated with 14 of the 15 thermal profiling datasets. However, closer inspection of dataset 4 revealed that only 17% of the quantified data reported $C_m$ values, because these were the only data that fitted well to a two-state unfolding curve. It thus follows that 83% of the data was disregarded because it did not fit to the two-state model (see Supplementary Table 1 for complete reference details). By contrast, between 66% and 99%, or 66% and 69% of proteins were reported to be well fitted to two-state unfolding isotherms in other thermal profiling and residue labeling datasets, respectively. Collectively, these findings suggested that the most canonical and simple patterns of unfolding, i.e., those that look like two-state unfolding curves were indeed the most likely to report on two-state-like protein unfolding behavior and to correlate with different apparent folding stability datasets. More intriguing however was that such two-state like stability values encompass only a fraction of the data available. Hence the remaining data likely was more applicably explained by complex unfolding mechanisms or mechanisms distinct to folding.

**Residue labeling techniques reveal nuanced and heterogenous changes in protein conformation due to chemical denaturation.** To further examine which data can be appropriately explained in terms of two-state folding or not, we collected our own dataset of apparent folding stability using tetraphenylethene maleimide (TPE-MI) as a probe for unfolded proteins. TPE-MI reacts with exposed cysteine free thiols that are otherwise buried in the folded state[6]. Free cysteine thiols are the least surface-exposed residue of all amino acids in globular proteins so provide an excellent target for examining protein foldedness[18]. We first performed a urea denaturation curve of purified β-lactoglobulin,

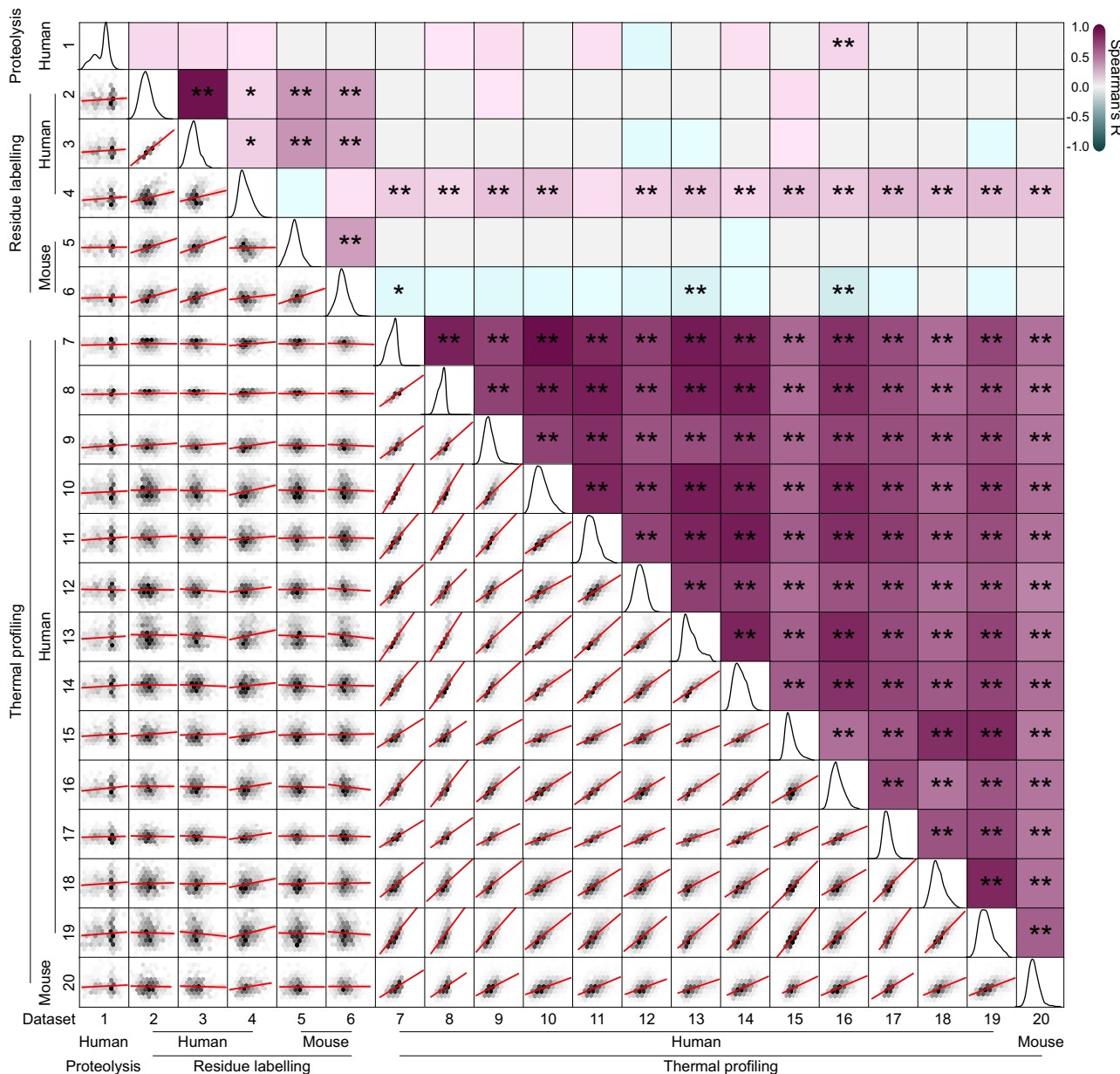

**Fig. 1 Limited correlations between measures of folding in published proteome stability datasets.** Shown are pairwise cross-correlations of normalized protein folding stability measures ($T_m$ or $C_m$). The bottom triangle shows hexbin density plots, where data point density tiles (grayscale) are overlaid with the fitted linear regression and corresponding 95% C.I. (red; in most cases intervals are too small to be seen). The diagonal shows kernel density distributions for individual datasets. The upper triangle shows pairwise Spearman's coefficients (R) represented in the form of a heatmap, overlaid with significance denoted by * ($p < 0.05$) or ** ($p < 0.01$). Exact $p$ values are provided in Supplementary Data 1. Datasets are ordered according to species of origin (human or mouse) and method of stability measure (limited proteolysis, residue labeling or thermal profiling). Datasets were derived from the following publications: Leuenberger et al., 2017, Science (dataset 1)[54], Ogburn et al., 2017, J Proteome Res. (datasets 2,3)[55], Walker et al., 2019, PNAS (dataset 4)[17], Roberts et al., 2016, J Proteome Res. (datasets 5,6)[56], Jarzab et al., 2020, Nat Methods. (datasets 7, 8, 9, 16, 17, 20)[45], Becher et al., 2016, Nat Chem Biol. (dataset 10)[57], Franken et al., 2015, Nat Protoc. (dataset 11)[58], Miettinen et al., 2018, EMBO J. (dataset 8)[59], Savitski et al., 2018, Cell. (datasets 13, 14)[10], Ball et al., 2020, Commun Biol. (dataset 15)[60], Savitski et al., 2014, Science. (dataset 18)[9], Sridharan et al., 2019, Nat Commun. (dataset 19)[15]. Complete reference information is provided in Supplementary Table 1. Source data are provided as a Source Data file.

which is a model globular protein containing a single buried free thiol residue, and unfolds via two-state-like behavior (Supplementary Fig. 1). The rate of reaction of TPE-MI with β-lactoglobulin was proportional to the anticipated exposure of the buried thiol upon two-state unfolding. The relationship between rate of reaction and urea concentration yielded a $C_m$ consistent with that obtained from intrinsic tryptophan fluorescence and in accordance with other published results on β-lactoglobulin folding[19].

To use TPE-MI on a proteome-wide scale we created denaturation curves of cell lysate with urea (Fig. 2a). Lysates were prepared from mouse neuroblastoma cells (Neuro2a) subjected to Stable Isotope Labeling by Amino acids in Cell culture (SILAC) using light or heavy ($^{13}C$ L-lysine and $^{13}C$,$^{15}N$ L-arginine) isotopes for quantitation of reactivity. In essence, lysate from light-isotope labeled cells was used as the "native" control versus lysate from heavy-isotope labeled cells prepared with different concentrations of urea. The light and heavy-isotope

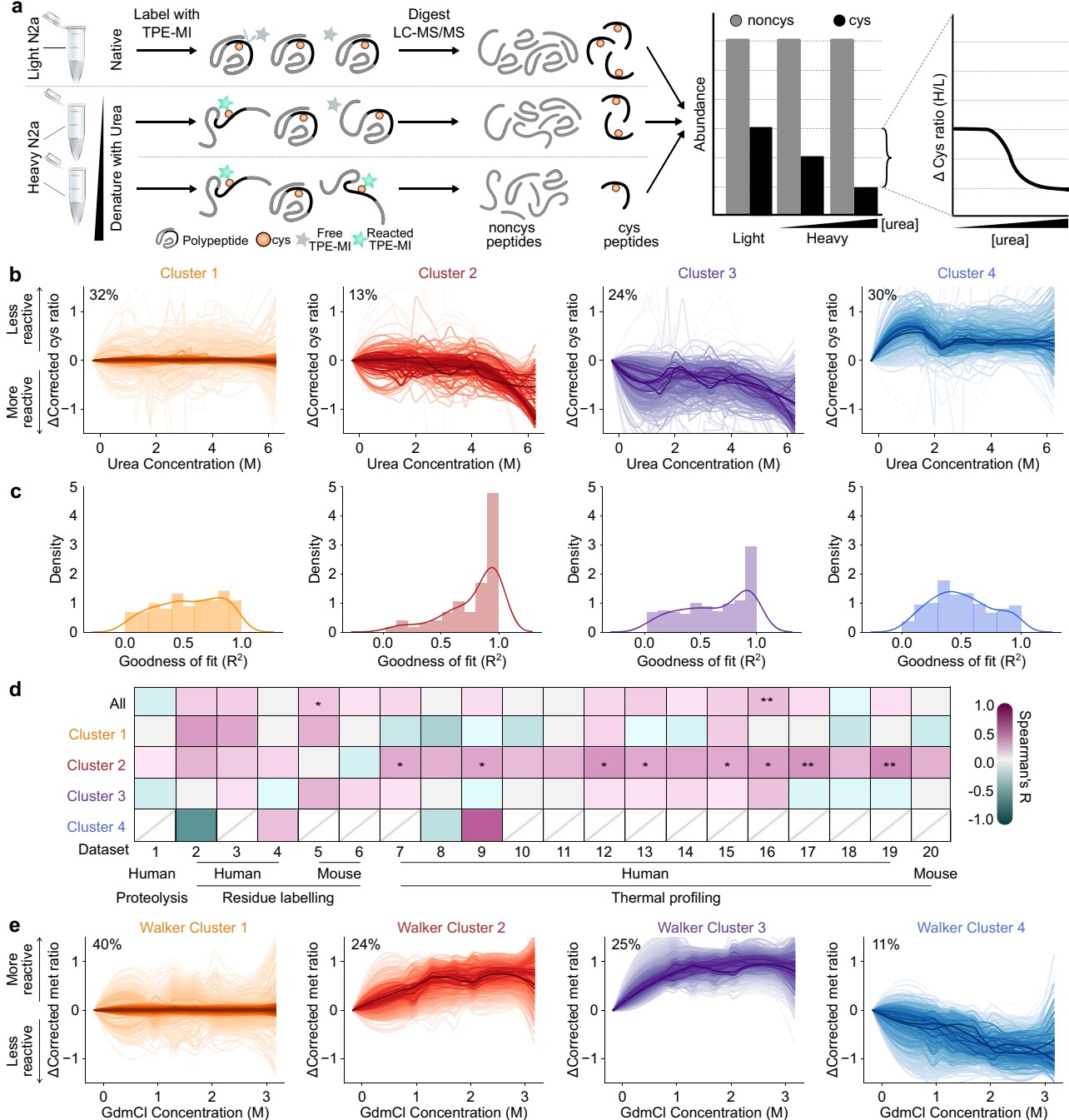

**Fig. 2 Residue labeling methods reveal patterns of change inconsistent with two-state unfolding during chemical denaturation. a** Schematic representation of the workflow used to detect conformational change using TPE-MI following chemical denaturation. The graphs show hypothetical data to illustrate the point. **b** Clustering of cysteine-peptide TPE-MI reactivity profiles as a function of urea concentration using fuzzy c-means. Families of individual peptide traces are shown where the color saturation reflects the cluster membership score for each peptide, such that the darkest traces are representative of the typical response for each cluster. Data ($n = 3$ biological replicates) were smoothed by significance scaling and loess smoothing. The proportion of total peptides assigned to each cluster is also indicated. **c** Histogram of goodness-of-fit among individual peptides following fitting to a two-state unfolding curve. **d** Cross-correlation analysis of peptides in each cluster to previously published stability datasets (list of datasets in Supplementary Table 1). Correlation heatmap is colored according to the Spearman's correlation coefficient ($R$) and significance is denoted by * ($p < 0.05$) or ** ($p < 0.01$). Exact $p$ values are provided in Supplementary Data 1. Datasets are ordered according to species of origin (human or mouse) and method of stability measure (limited proteolysis, residue labeling or thermal profiling). Missing points (gray line) indicate fewer than 5 proteins in common. **e** Corresponding clustering analysis of an independent published residue labeling dataset (dataset 4) that targeted oxidation of exposed Met residues[17]. Data here is represented as per panel B. Source data are provided as a Source Data file.

labeled samples were each reacted with TPE-MI before mixing and quantitation for the level of reactivity. Peptides modified with TPE-MI are not readily quantified via mass spectrometry due to unfavorable physicochemical properties. While we cannot distinguish TPE-MI from other covalent modifications of an individual cysteine, both report on a change in the local environment for the cysteine and result in a net loss of the unreacted peptide. Therefore, the extent of cysteine reactivity was determined from the change in abundances of peptides with unreacted cysteines normalized to the ratio of peptides from the same protein that lacked cysteine (Fig. 2a).

To determine the underlying trends in cysteine reactivity as a function of urea concentration we clustered the cysteine peptide reactivity profiles using an unbiased computational approach of fuzzy-c means[20,21]. This analysis yielded four distinct patterns of cysteine response to urea titration (clusters 1–4) shown in Fig. 2b. Cluster 1 appeared to represent patterns lacking a systematic change in reactivity with urea concentration. Clusters 2 and 3 both appeared to represent patterns of increasing cysteine thiol reactivity, which was anticipated for greater exposure of buried thiols upon denaturation. Cluster 2 differed from cluster 3 by appeared to show the reactivity changes occurring first at higher urea concentrations whereas the changes occurred at lower urea concentrations for cluster 3. Cluster 4, representing one third of the identified cysteine peptides, appeared to represent decreases in reactivity upon increasing concentrations of urea. This was counter to the anticipated increase in reactivity expected from the exposure of buried cysteine residues induced by denaturation, suggesting that distinct processes were occurring in some proteins in response to urea titration that led to greater burial of exposed cysteine thiols (discussed in more detail below).

To determine which of the data were most consistent with a two-state folding mechanism, individual cysteine reactivity curves were fitted to a two-state unfolding curve and assessed for goodness of fit (Fig. 2c). Consistent with the greater reactivity upon urea denaturation, clusters 2 and 3 contained the most peptides with good fits defined by an $R^2 > 0.9$ (Fig. 2c). However, the peptides in cluster 2 were the only group that showed a significant correlation between the fitted $C_m$ values with those of existing census datasets described herein previously in Fig. 1 (Fig. 2d). Given that the published datasets were pre-filtered in the original studies to be consistent with two-state unfolding, this finding demonstrates an authenticity in this subset of data for tracking bona fide two-state unfolding events. The peptides in cluster 3 did not share this correlation with other datasets, and may be a diagnostic of reactivity reporting on non-two-state folding mechanisms such as multistate or non-reversible folding, or other non-folding related mechanisms (Fig. 2d).

To investigate whether the general conclusions made from the TPE-MI dataset can be drawn in datasets derived from other residue labeling methods for foldedness, we re-examined the pre-processed peptide quantitation from dataset 4[17] according to our clustering procedures. Dataset 4 monitored methionine exposure in the lysate of human cell line, HCA2-hTert, by a free radical oxidation approach in different concentrations of the denaturant guanidine hydrochloride[17]. This dataset was therefore analogous to the TPE-MI approach but independent in multiple parameters of chemical denaturant (guanidine hydrochloride versus urea), species (human versus mouse), target residue for labeling (methionine versus cysteine) and research team (independent research labs). In addition, unlike TPE-MI, oxidized methionines are readily quantifiable, which offered us the opportunity to evaluate the applicability of our method to a directly measured modification.

Despite these differences in parameters, the clustering procedures resulted in a strikingly similar grouping of the oxidized methionine peptides to the TPE-MI dataset (Fig. 2e – note however that the direction of change is inverted due to the nature of the measurements) that illustrates several fundamentally consistent conclusions. First was that the data formed 4 clusters with similar patterns of response. Second was that there were broadly consistent proportions of peptides in the different clusters. Most notably was the consistent cluster for increased protection upon denaturant titration that indicated changes inconsistent with unfolding.

Together the TPE-MI and methionine oxidation data indicated that with an unbiased clustering procedure only about one quarter of residue labeling data in chemical denaturation curves of mammalian proteomes can be confidently described as consistent with a two-state unfolding mechanism. Most notably, this independent dataset confirmed the observation that a substantial portion of the reactivity changes measured in the presence of denaturant cannot be satisfactorily explained by unfolding.

**Residue labeling patterns earmark protein conformational rearrangement as well as unfolding.** To investigate protein properties that explain the different patterns of response to urea titration, we first examined the physicochemical characteristics of the peptides in each cluster (and the proteins to which they belong to; Fig. 3a). We examined properties predicted from amino-acid composition that pertained to likely burial of the target residues in the folded state, including charge, hydrophobicity, likelihood to reside in regions of secondary structure and relative solvent exposure for individual peptides associated with each cluster. Several characteristics stood out. Notably the peptides in clusters 2 and 3, which we have thus far demonstrated to have best consistency with two-state folding mechanisms, were most likely to contain hydrophobic, and least likely to be in unstructured protein regions which is in accordance with this conclusion (Fig. 3b). In addition, cluster 2 were most likely to contain solvent-buried residues. For the parent proteins that have known structures (around 20% of the proteins identified), the extent of solvent exposure of the labeled cysteine residue further supported the conclusion that cysteines in cluster 2 were in buried regions of proteins and hence became exposed upon denaturation (Supplementary Fig. 2a). For completeness of analysis, we did not identify any enrichment for other protein features, such as whether the peptides resided in active sites, binding sites, functional motifs, disulfide bonds or annotated domains (Supplementary Fig. 2b–d). Another feature we considered was the presence of conserved protein domains, annotated using the PFAM database which assigns protein domains to families according to conserved structural or sequence features that have been aligned using hidden Markov models. While the proportion of residues located within annotated PFAM domains was relatively high (more than 70% in every cluster), this was anticipated due to the likelihood of free thiols being buried in the folded state of most proteins[22].

Because individual proteins may include multiple cysteine-containing peptides that fall into different clusters, we next grouped proteins into categories based on which cluster the peptides belonged to (Fig. 3a). More than half of the peptides belonged to proteins that contained at least one other cysteine peptide from a different cluster (Fig. 3c). Proteins with such peptides were considered multi-clustered and separated from proteins that contained cysteine peptides exclusively in one of the other four clusters (which we hereon call uni-clustered) for further analysis. Of the multi-clustered proteins, one-third had at least one cysteine peptide which decreased in reactivity (i.e., was in cluster 4) and one or more cysteine peptides for which reactivity increased in the presence of higher concentrations of

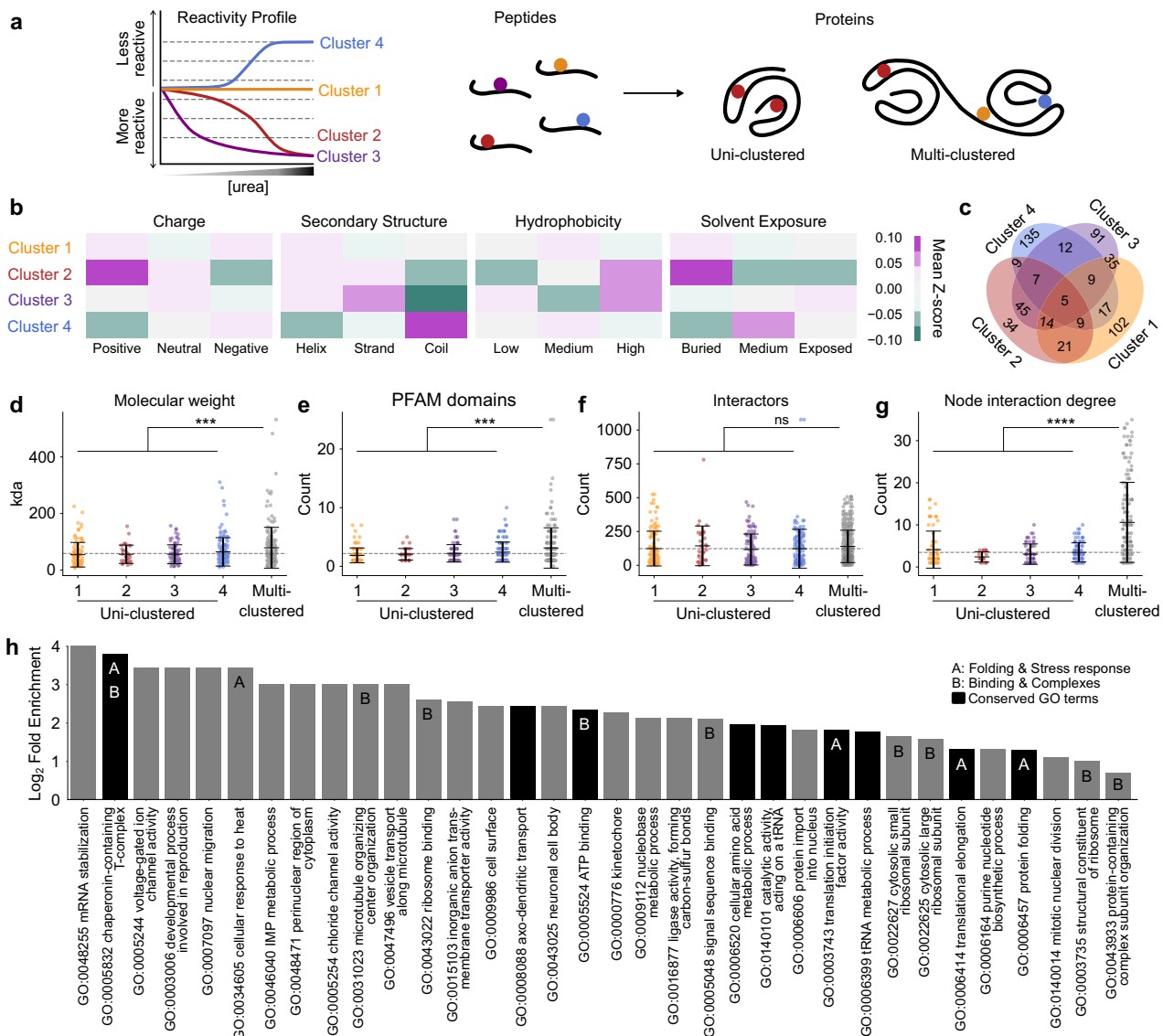

**Fig. 3 Functional responses to denaturation drive heterogeneous changes in reactivity within single proteins. a** Schematic overview of peptide cluster patterns and how proteins are grouped depending on the composition of cysteine peptides from multiple clusters. The left graph shows a schematic reactivity profile of each of the four clusters. Peptides are then assigned to their parent protein, which may be deemed uni-clustered if only consisting of cysteine-containing peptides from a single cluster, or multi-clustered where cysteine-containing peptides from a single protein are associated with more than one cluster. **b** Mean *z*-score for predicted/extracted physiochemical features according to peptide amino acid composition. **c** Venn diagram depicting proportion of proteins for which peptides were found in each cluster combination. **d** Molecular weight of and **e** number of annotated PFAM domains in proteins to which clustered peptides are assigned. **f** Number of high-confidence first-shell protein-protein interactions and **g** inter-cluster node interaction degree annotated in STRINGdb (v11.0, score >0.7) for proteins found in each cluster. **h** Gene ontology terms enriched among multi-clustered proteins. Enrichment was determined using Panther GOSlim Fisher's over-representation test with false-discovery rate correction. Common themes are denoted: A = protein folding and stress response, B = binding and complexes. Dark bars denote exact terms found to also be enriched among multi-clustered proteins in published dataset 4. Panels **d**–**g** show individual protein datapoints overlayed with mean ± S.D. Mean of combined uni-clustered proteins is shown as dotted gray line. Uni-clustered proteins were compared to those associated with multiple clusters via two-tailed *t*-test with Welch's correction, *** denotes $p < 0.001$, **** denotes $p < 0.0001$, ns denotes $p > 0.05$. Exact n and p values are provided in Supplementary Data 1. Source data are provided as a Source Data file.

urea, which suggested multimodal impacts on the protein during denaturation. It is reasonable to predict that such proteins are larger and multi-domain. Indeed, multi-clustered proteins were more likely to have a larger molecular mass and contain more annotated PFAM domains than uni-clustered proteins, consistent with this conclusion (Fig. 3d, e; two-sample *t*-test, $p < 0.001$). Also of note was the consistency in predicted physiochemical features for uni-clustered proteins whose peptides were associated with cluster 2, which featured elevated hydrophobicity and lower

solvent exposure compared to the other uni-clustered and multi-clustered categories (Supplementary Fig. 2e). This result further supported the conclusion that proteins containing solely cluster 2 peptides were the most likely to display two-state unfolding and that the other proteins with mixed clusters displayed more complex unfolding or other non-folding changes in response to urea titration.

In addition to physiochemical properties, we also examined the molecular functions of proteins assigned to the uni- and

multi-clustered protein categories. Protein-protein interaction analysis using the STRING database revealed no difference in the number of direct high-confidence protein binding partners between uni- versus multi-clustered proteins (Fig. 3f). However, the average node degree (the number of protein-protein interactions within each cluster) was up to four-fold higher in multi-clustered proteins than among uni-clustered proteins (Fig. 3g). This result was consistent with the anticipation that multi-clustered proteins are more likely to be multi-domained and larger in size. It therefore follows that such proteins would operate in larger functional networks, which display coordinated changes in cysteine reactivity where exposure of one protein of a complex yields exposure of its binding partner. By comparison, uni-clustered proteins, particularly those in cluster 2, were more likely to be poorly interconnected. This conclusion is consistent with an anticipation that these proteins were simpler globular proteins whereby the data reported solely on their foldedness and not their function in networks.

Gene ontology (GO) analysis was investigated for each of the protein categories to examine the possibility of a coordinated functional response corresponding to the cysteine reactivity changes (Fig. 3H). Of the 34 significantly enriched top-level GO terms in the multi-clustered proteins, half encompassed mechanisms pertaining to binding and protein complexes or proteostasis response mechanisms such as protein folding machinery. Of note, three of the GO terms (chaperonin-containing T-complex, GO:0005832; cellular response to heat, GO:0034605; and protein folding, GO:0006457) are associated with the stimulus of denaturation. These changes in thiol reactivity appeared to earmark changes in ligand binding or the conformation of select proteins that have functions in responding to unfolded proteins that accumulate at higher concentrations of urea. A parallel analysis of the methionine oxidation data (dataset 4) yielded nine identical GO terms (Fig. 3h for common terms, Supplementary Fig. 3 for all dataset 4 specific terms). This result was further striking in that data from a distinct method, species and denaturant led to a conserved GO enrichment of terms related to responses to unfolded proteins, including identical terms of chaperonin-containing T-complex (GO:0005832) and protein folding (GO:0006457). These data therefore led us to conclude that multi-clustered proteins were likely to encompass functional responses to protein denaturation. By contrast, GO analysis of uni-clustered proteins showed no conserved terms between the two datasets (Supplementary Fig. 4).

**Residue reactivity captures features of chaperone conformational change in lysate.** To decipher the molecular mechanisms that underlie the functional responses to unfolded protein, we next focused on a class of proteins that we expected to have functions in engaging with unfolded proteins, i.e., molecular chaperones. 42 proteins annotated with the ontology term "chaperone-mediated protein folding" machinery (GO:0061077) were identified across the protein groups, particularly in the multi-clustered proteins (Supplementary Fig. 5).

One of these proteins HSPA8 (HSC70; P63017) is the cognate heat shock protein 70 (Hsp70), and was identified by 3 cysteine peptides assigned to two different clusters (Fig. 4a). HSPA8 binds to unfolded proteins in concert with J-domain protein co-chaperones and nucleotide exchange factors[14]. Together they drive protein folding through ATP-dependent cyclical binding and release[23]. A conserved structural feature of Hsp70 proteins are four modules: an N-terminal nucleotide binding domain (NBD), a substrate binding domain (SBDβ), a helical lid domain (SBDα), and a disordered C-terminal tail of variable length[24]. The disordered tail of HSPA8 comprises an EEVD motif that mediates interactions with cofactors such as J-domain protein DNAJB1.

One peptide from the NBD, containing Cys17, became more reactive to TPE-MI at urea concentrations greater than 4 M (Fig. 4d). The other two peptides were from the SBDα domain, containing Cys574 and Cys603, and these both became more protected in concentrations of urea between 3 and 5 M (Fig. 4d). We also identified one peptide from DNAJB1, that contained two cysteine residues (Cys267, Cys269). These cysteines displayed a complex biphasic (but overall increased) reactivity profile upon exposure to increasing concentrations of urea (Fig. 4d). We therefore postulated that the changes in cysteine reactivity in HSPA8 and possibly cofactor DNAJB1 upon urea titration may arise due to allosteric conformational changes resulting from their binding to each other and/or upon the engagement with unfolded proteins.

To test this hypothesis, we examined thiol reactivity changes in a reconstituted HSPA8 system in vitro that comprised of purified human HSPA8, DNAJB1 and a well characterized model client, malate dehydrogenase (MDH2)[25] (Fig. 4a–c). To look specifically for changes resulting from interaction with denatured client, TPE-MI reactivity was compared between reconstituted systems containing native versus thermally denatured MDH2 using tandem mass tag (TMT) isotopic labeling (Fig. 4e).

First, we assessed the system without added ATP. Under this condition HSPA8 and DNAJB1 can bind to non-native MDH2 substrate to form a stable complex[26,27]. More specifically, the SBDα domain of an Hsp70 protein (DnaK) has been shown to interact with substrate when the chaperone is in the ADP-bound state[28]. We saw no change in reactivity of the HSPA8 NBD peptide containing Cys17 (Fig. 4f) suggesting that the increase in reactivity observed above 4 M urea titration in lysate was attributable to NBD unfolding. In contrast, the SBDα domain peptide containing Cys574 decreased in reactivity (one-sample $t$-test, $p = 0.033$). This result therefore suggested that the decrease in reactivity observed in the SBDα region during urea denaturation arose from substrate and/or ligand binding.

In DNAJB1, we observed two cysteine peptides in the reconstituted system. One peptide, containing Cys179, is located close to the hinge between two β-barrel-like subdomains (CTDI and CTDII) that binds to substrates (Fig. 4a). The second, containing two closely adjacent cysteines (Cys267, Cys269), is in the homodimer interface. It is important to note that we could not ascribe reactivity changes to a single cysteine within this peptide, thus the changes represent an average across these residues. The peptide containing Cys267 and Cys269 became significantly more reactive under these conditions (Fig. 4f; one-sample $t$-test, $p = 0.023$). The central location of this peptide within the homodimeric interface of DNAJB1 suggested that accumulated client-binding altered the conformation of DNAJB1 to expose the structure nearby these cysteines.

Next we examined the effect of adding supplemental ATP to the reconstituted system, which we predicted would fuel HSPA8 to undergo the full catalytic cycle and therefore release accumulated complexes of HSPA8 and DNAJB1 bound to client[24]. The Cys574 peptide from HSPA8 became more reactive under these conditions, which is in agreement with HSPA8 disengaging client and/or DNAJB1. Intriguingly, the Cys179 peptide from DNAJB1 showed increased reactivity (Fig. 4f; one-sample $t$-test, $p = 0.023$). Of note, this peptide is close to the region of DNAJB1 shown to bind the EEVD motif of HSPA8[29], suggesting a level of deprotection when client-bound complexes of the HSPA8-DNAJB1 machinery are dissociated. While this is an attractive hypothesis, we could not exclude the possibility of other allosteric changes associated with DNAJB1 activity. Namely, intramolecular J-domain interactions with the hinge region occur near this residue which are modulated by DNAJB1 engagement with HSPA8[25,30–32]. The peptide in DNAJB1

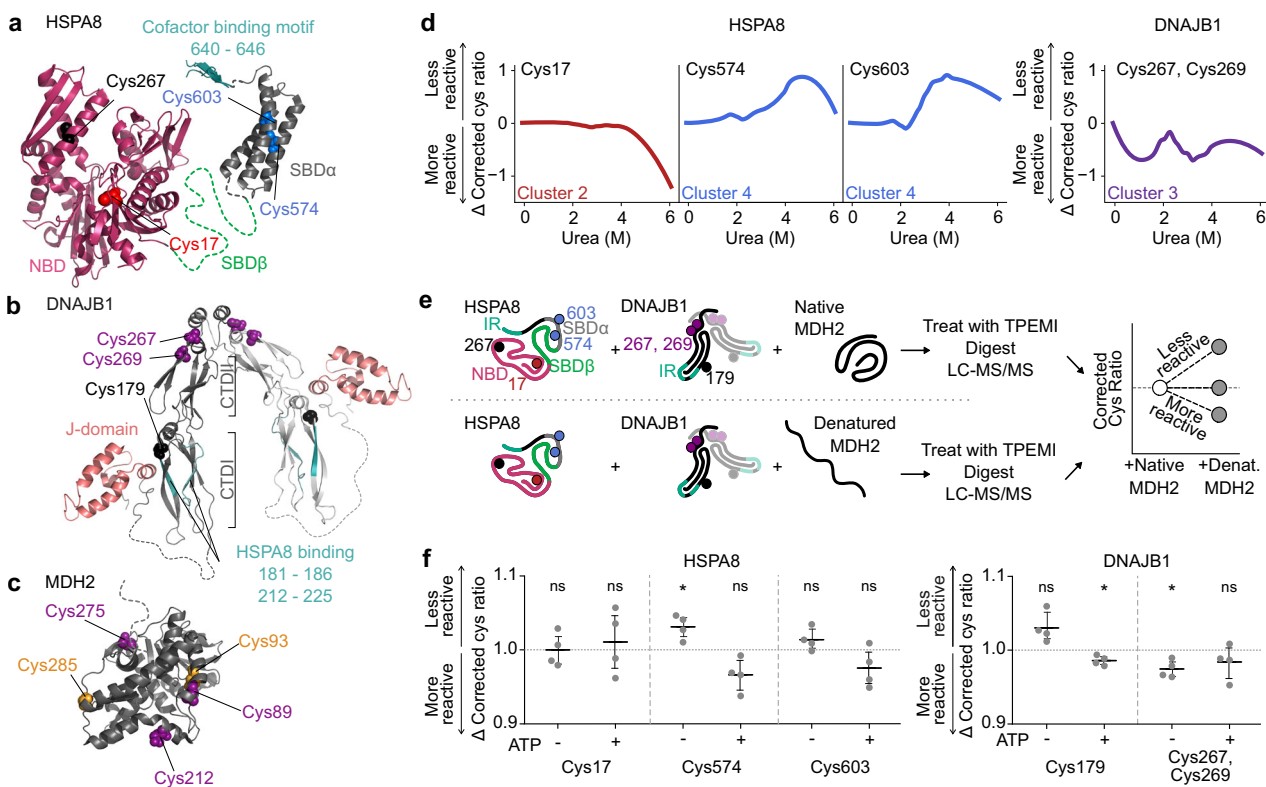

**Fig. 4 Detection of distinct HSPA8 conformations by residue labeling approaches. a** Ribbon structure of HSPA8 (models built with PDB 4H5R[61], 3AGY[29] and 4KBQ[62]). Nucleotide binding domain (NBD; ruby), substrate binding domains (SBDβ green, SBDα dark-gray) and cofactor binding motif (EEVD motif; teal) are shown on protein backbones. **b** Ribbon structure of DNAJB1 (model built with PDB 3AGZ[29] and 1HDJ[63]), with dimer composed of two monomers colored dark and light gray respectively. J-domain (salmon) and HSPA8 binding region (teal) are shown on protein backbones, and C-terminal domains (CTDI and CTDII) are indicated by brackets. **c** Ribbon structure of MDH2 (PDB 1MLD[64]). Dotted lines represent sequence regions with missing structural information **d** Changes in thiol reactivity of HSPA8 and DNAJB1 peptides in Neuro2a lysate titrated with urea. **e** Schematic for recombinant client-binding assay. Human HSPA8 and DNAJB1 were incubated with native (above line) or heat-denatured (below line) client, MDH2. Nucleotide binding domain (NBD; ruby), substrate binding domains (SBDβ; green and SBDα; gray) and cofactor interaction region (IR; teal) are shown on protein backbones. In the case of DNAJB1, dimer is shown with second monomer desaturated. After incubation, samples were labeled with TPE-MI then prepared for mass spectrometry. Finally, the corrected cys ratio for each of the HSPA8 and DNAJB1 was compared between samples containing native and denatured client, where the corrected cys ratio will increase for those residues that become less reactive. **f** Change in cysteine reactivity of peptides derived from native human HSPA8 and DNAJB1 when incubated with heat-denatured MDH2, in the absence or presence of exogenous ATP prior to TPE-MI labeling. Means ± S.D. ($n = 4$ biological replicates) are shown, and deviations from the expected mean of 1 were tested using a one-sample $t$-test (* denotes $p < 0.05$, ns denotes $p > 0.05$). Exact $p$ values are provided in Supplementary Data 1. In all panels, cysteine residues are labeled and colored according to the cluster their respective peptides were assigned (orange, red, purple and blue correspond to clusters 1–4 respectively, black was not observed). Source data are provided as a Source Data file.

containing (Cys267, Cys269) was no longer exposed (Fig. 4f; one-sample $t$-test, $p = 0.26$), which is consistent with the lack of accumulated substrate-chaperone complexes that would otherwise drive the conformational changes at the dimer interface described above.

We also examined the reconstituted chaperone experiment under conditions of urea denaturation to mimic the lysate denaturation experiment (Supplementary Fig. 6a). In this context, Cys17 in HSPA8 showed no change in reactivity (Supplementary Fig. 6a; one-sample $t$-test, $p = 0.14$ and $p = 0.68$), which is consistent with the NBD being denatured and hence making Cys17 amenable to reaction irrespective of other ligands. This conclusion is supported by a recent study reporting a urea denaturation curve of HSPA8 that showed it displayed two-domain folding behavior[33]. One domain, likely the NBD, had a $C_m$ around 1.5 M urea and the other domain a $C_m$ around 3.6 M urea. The changes in the SBD also showed protection at Cys603 that was abolished by the addition of ATP (Supplementary Fig. 6a; one-sample $t$-test, $p = 0.044$ and $p = 0.31$ respectively). Hence this is consistent with residual binding activity in the

presence of urea that protect the cys from reaction. DNAJB1 appeared to be mostly denatured at 6 M urea (Supplementary Fig. 6c) and accordingly the peptide Cys179 did not show any change in reactivity (Supplementary Fig. 6a; one-sample $t$-test, $p = 0.22$ and $p = 0.12$ in the absence or presence of ATP respectively). The DNAJB1 peptide containing Cys267 and Cys269 showed no change in reactivity in the absence of ATP (Supplementary Fig. 6a; one-sample $t$-test, $p = 0.15$) but there was a significant protection in the presence of ATP (Supplementary Fig. 6a; one-sample $t$-test, $p = 0.015$). This protection was surprising and there is no simple interpretation but given the changes we observed in these regions of DNAJB1 with the thermally denatured MDH1, the results may underly poorly understood conformational changes in that region of DNAJB1.

To determine which, if any, protein-protein interactions may be responsible for the protection we observe in HSPA8 in lysate, we repeated the urea denaturation curve of lysate with cells grown in the absence and presence of the specific HSP70 inhibitor, VER155008 (Fig. 5a). In this experiment we compared the individual peptide abundances from control-derived lysate to

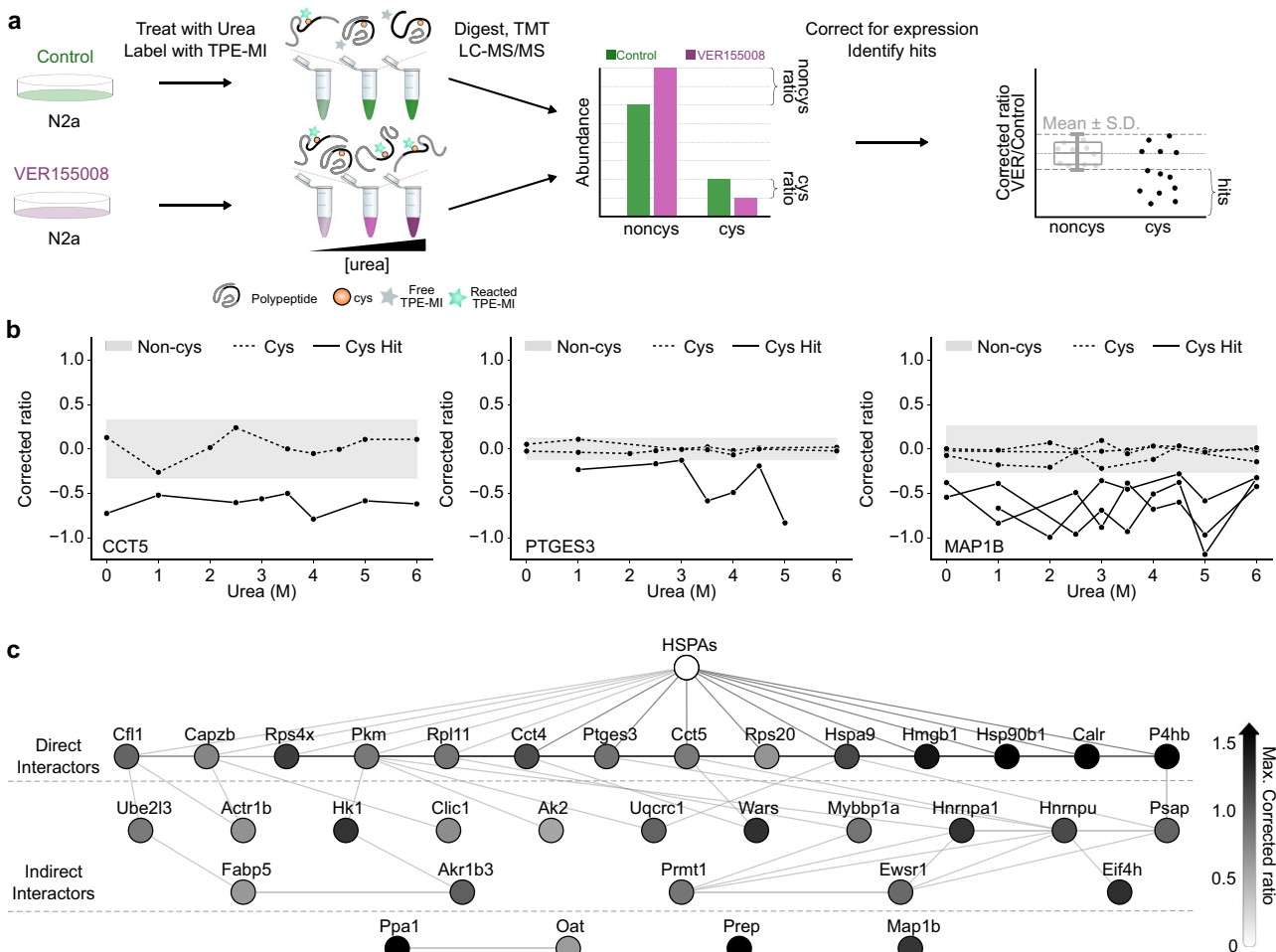

**Fig. 5 Inhibition of HSP70s alters the TPE-MI reactivity of known HSP70 family interactors. a** Schematic representation of the workflow used to detect conformational change using TPE-MI following chemical denaturation. Hit proteins whose denaturation cys reactivity profile was altered by VER15008 treatment were identified as those containing at least one peptide whose deviation exceeded the standard deviation of the corrected per-protein non-cysteine peptide ratio. The graphs show hypothetical data to illustrate the point **b** Exemplar peptide traces for hit proteins CCT5 (P80316), PTGES3 (Q9R0Q7) and MAP1B (P14873). Gray band represents the cumulative S.D. of non-cysteine peptides across all urea concentrations. **c** Protein interaction network for hit proteins, alongside major HSP70 isoforms HSPA1A, HSPA1B, and HSPA8 (HSPAs). Individual nodes are colored according to the absolute maximum corrected ratio across all quantified cysteine-containing peptides for that protein. Interactions were extracted from STRINGdb (v11.0, medium confidence score >0.4) and manually grouped according to direct (edge distance = 1) and indirect (edge distance >1) interaction with an HSP70. Source data are provided as a Source Data file.

VER155008-derived lysate at matching concentrations of urea, such that any peptides whose reactivity was not affected by the presence of VER155008 would have a ratio of 1. After correcting for any differences in expression of individual proteins using the mean ratio for non-cysteine-containing peptides for each protein, we determined potential hit cysteine peptides as those whose corrected ratio exceeded the variability of the non-cysteine-containing peptides across all denaturation concentrations for a given protein (Fig. 5a). 34 proteins were found to have a significant hit in cys-reactivity, examples of which included T-complex protein Ring Complex (TRiC) subunit CCT5 (P80316), the molecular chaperone prostaglandin E synthase 3 (PTGES3; Q9R0Q7) and microtubule-associated protein 1B (MAP1B; P14873) (Fig. 5b). Both CCT5 and PTGES3 share a known interaction with HSP70 family members and each had one peptide that was more reactive in the presence of the HSP70 inhibitor, consistent with these regions being protected by HSP70 during denaturation. Protein interaction analysis of the other proteins identified, alongside HSP70 family members HSPA1A (Q61696), HSPA1B (P17879) and HSPA8 (P63017), revealed

them to be enriched as known HSP70 interactors (Fisher's exact test, $p = 0.031$). 14 of the proteins were found to have a direct interaction with HSP70 and a further 16 were connected to an HSP70 isoform by one or more indirect interaction partners (Fig. 5c). These findings point to cys-reactivity changes arising through a functional network associated with Hsp70 function upon urea denaturation.

Last, we examined the other chaperone proteins in cluster 4 for clues on hidden patterns related to substrate binding. Major chaperone classes were represented in cluster 4, including the Hsp40, Hsp70 and Hsp90 families, the Bcl-2 associated athanogene cochaperone (BAG) family and the T-complex protein Ring Complex (TRiC) chaperonin. Many of the chaperones showed decreased cys reactivity in protein sequences previously associated with chaperone function (Supplementary Table 2). This includes Cys530 of HSP90AA1 (Hsp90α; P07901), which is located in the 'middle domain' and is spatially close to residue cluster (E528, Y529, Q532) required for client binding and chaperone function (Fig. 6a)[34]. Thirty-three unique cysteine-containing peptides were observed across all eight subunits of

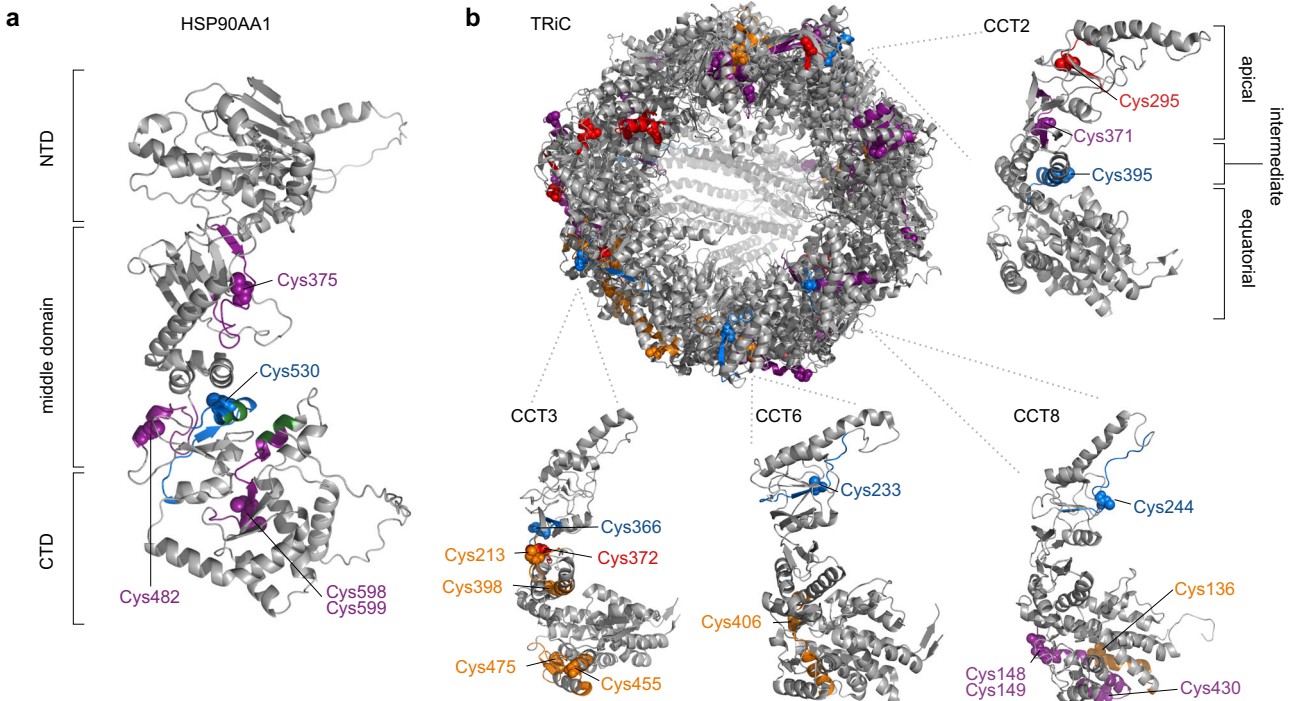

**Fig. 6 Regions associated with chaperone function are protected during denaturation. a** Ribbon structure of HSP90AA1 (P07901) as predicted by Alphafold v1[65]. Brackets delineated the N-terminal (NTD; responsible for nucleotide binding), middle (MD), and C-terminal (CTD; responsible for dimerization) domains. **b** Ribbon structure of human T-complex protein Ring Complex (TRiC) derived from PDB 6NRC[66]. Peptides were mapped from Mus musculus to Homo sapiens using blastp alignment. Zoomed and rotated views are also provided for subunits of interest CCT2 (P78371), CCT3 (P49368), CCT6 (P04227) and CCT8 (P50990). Common domain structure comprising the apical domain (substrate binding and lid), intermediate domain, and equatorial domain (ATP-binding) is delineated on CCT2. In all panels, cysteine residues are labeled and colored according to the cluster their respective peptides were assigned (orange, red, purple and blue correspond to clusters 1–4 respectively, black was not observed). Additional residues of interest in substrate binding are colored in green.

TRiC (TCP1; CCT2-CCT8). Of note was that four peptides derived from CCT2, CCT3, CCT6 and CCT8 had decreased reactivity at high urea concentrations (Fig. 6b).These peptides correspond to the apical or intermediate-equatorial hinge regions that are known to bind to substrate, or have conformational changes associated with substrate binding[35,36]. These chaperones provide a wealth of candidates with well-characterized structure-function relationships for further investigation of the denaturation-induced protection phenomena.

## Discussion

Collectively, our findings provide a means to reveal hidden complexity in proteome-wide datasets targeting foldedness with residue labeling approaches. Namely, we demonstrate that a substantial component of the changes seen in residue labeling datasets applied to study proteome denaturation by chemical denaturants are better explained by changes in protein conformation and ligand interactions than unfolding. These findings have two implications of note. First is that analysis of the peptides (and proteins) that generate denaturation curves most similar to two-state unfolding curves provides the most consistent correlation between methodological approaches and provide a more robust core census list of stability measurements. While the absolute stability measurements may not be important if they are used as internal reference points for the effect of a particular treatment such as drug binding, their use for in situ absolute measures of folding stability needs to be carefully considered in any retrospective analysis of whole proteome stability datasets because it highlights that upwards of two thirds of the data previously fitted to a measure of folding stability may be more

appropriately explained by changes in conformation or ligand association. Other proteomics approaches have drawn general conclusions that are in agreement with our findings here, namely how changes in proteome solubility encode information on rewired protein interaction networks[37–39]. Others have also shown that ligand interactions can modulate the thermal melting profiles of proteins in lysate[8,40,41], and recent commentary speculates this is one among a range of biophysical effects that could contribute to a protein's observed thermal stability[42]. The workflow presented here provides a useful strategy to delineate changes arising from unfolding from these other attributes of proteins.

The second implication is that the ability to determine subtle changes in tertiary and quaternary conformation with domain-level resolution distinguishes amino-acid specific methods from thermal melt and aggregation-based techniques. As such the approaches extend beyond other approaches, especially limited proteolysis, for such domain-level resolution of conformational changes[16]. Subtle changes in protein conformation mediate protein-protein interactions underlying many cellular functions. However, the dynamic and often transient nature of these interactions can make them challenging to quantify in cells en masse and requires sensitive but non-specific conformational probes capable of distinguishing domain-specific changes. Overall, the data presented here support the ability of residue labeling methodologies such as TPE-MI to fill this void, providing quantitative insight into aspects of protein conformation beyond stability and unfolding. Like all residue labeling approaches, we acknowledge there are potential caveats with chemical labeling proteins such as perturbation of protein structure upon reaction. Such modifications have the potential to affect long-range

conformational rearrangements[43,44]. And with the TPE-MI method the absence of directly-quantifiable modified peptides means we must infer modification based on loss of the non-modified form. However, one big advantage is the capacity to probe protein structures in live cells, which we anticipate to open the door for studies on proteome structure and function in natural intact biological settings.

## Methods

**Materials**. All materials were purchased from Sigma-Aldrich (St. Louis, MO, USA) unless otherwise indicated. The mouse neuroblastoma cell line Neuro2a (N2a) was obtained from lab cultures originating from the American Type Culture Collection and screened for mycoplasma contamination. TPE-MI was stored as stocks (10 mM in DMSO) in the dark at 4 °C before use. Recombinant human HSPA8 and DNAJB1 were purified as previously described[25]. Briefly, DNAJB1 and HSPA1A were expressed using the pCA528 vector with a 6×His-Smt3 tag in the BL21(DE3)/pRARE *Escherichia coli*. For inducing protein expression, 0.5 mM isopropyl-1-thio-D-galactopyranoside (IPTG, Sigma-Aldrich) was added and the cells were cultured further for 3 h at 30 °C. Cells were harvested by pelleting and lysed in 50 mM HEPES-KOH, pH 7.5, 750 mM KCl, 5 mM MgCl₂, protease inhibitor cocktail (Roche), 2 mM phenylmethylsulphonyl fluoride and 10% v/v glycerol, and 50 mM HEPES-KOH, pH 7.5, 300 mM KCl, 5 mM MgCl₂, protease inhibitor cocktail, 2 mM phenylmethylsulphonyl fluoride and 10% v/v glycerol for DNAJB1 and HSPA1A, respectively. Lysates were cleared by centrifugation at 30,000 × g (30 min, 4 °C) and the resulting supernatants were applied to a Ni-IDA matrix (Protino) and incubated for 60 min at 4 °C. Subsequent washing steps were performed with high and low salt buffers (50 mM HEPES-KOH, pH 7.5, 750 and 500 mM KCl, 5 mM MgCl₂ and 10% v/v glycerol for DNAJB1; 50 mM HEPES-KOH, pH 7.5, 300 and 100 mM KCl, 5 mM MgCl₂ and 10% v/v glycerol for HSPA1A). An additional ATP wash (10 mM ATP (pH 7.0) in 50 mM HEPES-KOH, pH 7.4, 100 mM KCl, 5 mM MgCl2, 1 mM β-mercaptoethanol, 10% v/v glycerol) was performed. Protein was eluted with 300 mM imidazole in the corresponding low salt buffers. Dialysis was performed overnight at 4 °C in the presence of 4 mg His-tagged Ulp1 per mg substrate protein for proteolytic cleavage of the 6×His-Smt3 tag. The 6×His-Smt3 tag and His-Ulp1 were removed by incubating the dialyzed proteins in Ni-IDA matrix for 60 min at 4 °C and the proteins were then further purified by gel filtration chromatography using a Superdex 200 column (GE Healthcare).

**Correlation of published proteome stability datasets**. Published basal proteome stability datasets were collected as follows. First 813 articles were collected from PubMed keyword searches performed on July 31st 2020 for "thermal proteome profiling", "thermal proteome unfolding", "folding stability proteome", "limited proteolysis proteome", "proteome denaturation labeling", "proteome unfolding label" and "SPROX proteome". Abstracts were filtered manually for those which (i) reported primary experimental data for proteome-wide protein stability under control conditions, (ii) were derived from either human or mouse samples, and (iii) provided a publicly-accessible summary of the dataset. A total of 12 papers were found to meet these criteria. Datasets were then assigned into one of three categories based on methodology: limited proteolysis, residue labeling and thermal profiling. Complete details for the selected resources, including specific supplementary materials files for each dataset, are provided in Supplementary Table 1.

Datasets were collected from the relevant supplementary materials analyzed with custom scripts written in Python programming language. The logic of the scripts was to collect the reported protein stabilities provided by each source, and where necessary map the protein identifiers to UniProt Accession numbers. Proteins were mapped to KEGG Orthology (KO) identifiers via cross-referencing[45] from UniProt [https://www.uniprot.org/]. Stability measures of different datasets were filtered as per the goodness-of-fit criteria used in the original study, then normalized to 1 to enable cross correlation (i.e., to account for different scales for thermal denaturation values ($T_m$) and chemical denaturation values ($C_m$)). Spearman's correlation coefficients and *p* values were calculated in a pairwise manner for all proteins found to be commonly quantified in a given pair of datasets.

**Recombinant β-lactoglobulin denaturation**. A stock solution of recombinant β-lactoglobulin was prepared in PBS (pH 7.4), before being diluted to a final concentration of 250 µM in urea prepared at concentrations ranging from 0 to 6 M. Samples were then equilibrated in denaturant for 4 h at 25 °C, before being labeled with TPE-MI (or the vehicle control DMSO) at a final concentration of 50 µM[6]. Immediately after addition of the labeling reagent, samples were transferred to clear-bottom 96-well UVStar plate (Grenier BioOne). TPE-MI (350/20 nm ex, 465/20 nm em) and intrinsic tryptophan (295/10 nm ex, 360/20 nm em) fluorescence was read every 60 s for 60 min using a CLARIOstar (BMG Labtech) with shaking at 200 rpm for 5 s prior to each cycle. In the case of TPE-MI, the first 9 min of linear increase in fluorescence were fitted with linear regression to derive the rate of reaction of TPE-MI (gradient of the regression). Subsequent values were then normalized to the maximum gradient across all samples, before being fit via non-linear least squares regression to a two-state unfolding curve. In the case of

intrinsic tryptophan fluorescence, the raw fluorescence intensity values at the assay endpoint were normalized to the maximal intensity across all samples, then fit via non-linear least squares regression to a two-state unfolding model. This model accounts for the pre- and post-folding transition baselines[46], which are underlying linear relationships commonly seen at extreme denaturant concentrations among experimental measures of foldedness. These baselines are unrelated to the actual process of folding, and thus must be extrapolated to evaluate the equilibrium constant within the transition region. The following equation, which accounts for these states, was used to fit denaturation curves:

$$y = \frac{(y_f + m_f\,[D]) + (y_u + m_u\,[D]) \times e^{\frac{-m(C_m - [D])}{RT}}}{1 - e^{\frac{-m(C_m - [D])}{RT}}} \qquad (1)$$

where $y_f$, $m_f$, $y_u$ and $m_u$ correspond to the intercept and slope of the pre- and post-transition states respectively, $[D]$ corresponds to denaturant concentration, $m$ is the fitted slope of the transition region, and $C_m$ is the concentration at which 50% of the protein is unfolded. $T$ and $R$ correspond to the temperature (298.15 K) and gas constants, respectively.

**Cell culture**. Neuro2a cells were cultured in Dulbecco's modified Eagle's medium (DMEM; ThermoFisher Scientific) supplemented with 10% (v/v) fetal bovine serum (ThermoFisher Scientific) and 1 mM L-glutamine (ThermoFisher Scientific). In the case of isotopically labeled cultures (SILAC), cells were cultured in DMEM (Silantes) supplemented with either unlabeled (light) or ¹³C L-lysine and ¹³C,¹⁵N L-arginine, along with 10% (v/v) dialyzed fetal bovine serum (ThermoFisher Scientific) and 1 mM L-glutamine (Silantes). Cells were cultured in isotopically labeled media for at least 8 doublings prior to use. Cells were maintained at 37 °C in a humidified incubator with 5% atmospheric CO₂ and were reseeded into fresh culture flasks once at 80% confluency following mechanical dissociation. For plating, cell count and viability were automatically determined using a Countess trypan blue assay (ThermoFisher Scientific). Cells were seeded in 6 or 12 well plates (Corning) and grown for at least 18 h before treatment.

**Lysate preparation and chemical denaturation**. Following treatment, cells were washed once in PBS before being mechanically harvested in fresh PBS and centrifuged at 150 g for 5 min. Cell pellets were then resuspended in lysis buffer (50 mM Tris, pH 8.0, 0.8 % (v/v) IGEPAL CA-630, 1.5 mM MgCl₂) containing cOmplete EDTA, EDTA-free Protease Inhibitor Cocktail (Sigma) and 250 U benzonase (Sigma), then incubated on ice for 30 min. Lysate was then centrifuged at 20,000 g for 10 min to pellet cell debris, and the resultant supernatant transferred to a fresh Eppendorf tube. Total protein concentration was then determined using a Pierce BCA protein assay (Thermo Scientific) with bovine-serum albumin as the mass standard. A standard volume of lysate was distributed to aliquots of urea prepared at concentrations ranging from 0 to 6 M (in increments of 0.5 M) in water from an 8 M stock for which the concentration was determined by measuring the refractive index. In the case of SILAC lysate, light and heavy-labeled samples were combined at a 1:1 ratio prior to denaturation. Samples were then equilibrated in denaturant for 4 h at 25 °C. Following denaturation, lysate aliquots were labeled with TPE-MI to a final concentration of 100 µM (lowered from 200 µM used for labeling intact cells[6] to decrease the risk of inducing unfolding in TPE-MI saturated proteins) for 30 min at 25 °C, then immediately transferred to a 5-fold excess (v/v) of ice-cold acetone and stored at –20 °C overnight.

**Sample preparation for mass spectrometry**. Samples were pelleted at 20,000 g for 30 min at 4 °C. Protein pellets were solubilized in 100 µl of 8 M urea in 50 mM triethylammonium bicarbonate (TEAB), and incubated with shaking at 37 °C for 45 min. Proteins were reduced using 10 mM tris(2-carboxyethyl)phosphine, pH 8.0, and alkylated with 10 mM iodoacetamide for 45 min, before being digested with 2 µg trypsin (ThermoFisher Scientific) overnight with shaking at 37 °C. Peptides were then desalted by solid-phase extraction using an Oasis HLB 1 cc Vac Cartridge (catalog number 186000383, Waters Corp., USA) that was pre-equilibrated by washing. Samples were collected in fresh tubes and lyophilized (VirTis Freeze Dryer, SP Scientific). The lyophilized peptides were subjected to another round of BCA assay as above so as to normalize for similar loading onto the mass spectrometer. The final concentration of peptides was 0.1 µg/µl in 2% (v/v) ACN containing 0.05% (v/v) trifluoroacetic acid.

**NanoESI-LC-MS/MS**. Samples were analyzed by nanoESI-LC-MS/MS using a Orbitrap Fusion Lumos Tribrid mass spectrometer (Thermo Scientific) fitted with a nanoflow reversed-phase-HPLC (Ultimate 3000 RSLC, Dionex). The nano-LC system was equipped with an Acclaim Pepmap nano-trap column (Dionex—C18, 100 Å, 75 µm × 2 cm) and an Acclaim Pepmap RSLC analytical column (Dionex—C18, 100 Å, 75 µm × 50 cm). For each LC-MS/MS experiment, 0.6 µg of the peptide mix was loaded onto the enrichment (trap) column at an isocratic flow of 5 µl min⁻¹ of 3% ACN containing 0.1% (v/v) formic acid for 5 min before the enrichment column was switched in-line with the analytical column. The eluents used for the LC were 0.1% (v/v) formic acid (solvent A) and 100% ACN/0.1% formic acid (v/v) (solvent B). The gradient used (300 nl min⁻¹) was from 3–22% B in 90 min, 22–40% B in 10 min and 40–80% B in 5 min then maintained for 5 min before re-equilibration for 8 min at 3%

B prior to the next analysis. All spectra were acquired in positive ionization mode with full scan MS acquired from $m/z$ 400–1500 in the FT mode at a mass resolving power of 120,000 after accumulating to an AGC target value of 5.00 $e^5$, with a maximum accumulation time of 50 ms. Lockmass of 445.12002 was used. Data-dependent HCD MS/MS of charge states >1 was performed using a 3 s scan method, at an AGC target value of 1.00 $e^4$, a maximum accumulation time of 54 ms, a normalized collision energy of 35%, and with spectra acquired at a 7500 mass resolving power of 15,000. Dynamic exclusion was used for 45 s.

In the case of TMT-labeled samples, data were obtained on an Orbitrap Eclipse Tribrid mass spectrometer using nanoESI-LC parameters as described above. All spectra were acquired in positive mode with full scan mode full scan MS from $m/z$ 300–1600 in the FT mode at 120,000 mass resolving power after accumulating to a target value 5.00 $e^5$ and with maximum accumulation time of 50 ms. Lockmass of 445.12002 was used. A preferred inclusion list containing doubly and triply masses of tryptic peptides belonging to HSPA8 (P11142), DNAJB1 (P25685) and MDH2 (P00346) was created to increase the coverage of identifiable peptides from these proteins. Data-dependent HCD MS/MS of precursors that matches the inclusion list then other charge states >1 were performed using a 3 s scan method, 0.7 $m/z$ isolation width, target value of 5.00 $e^4$, a maximum accumulation time of 54 ms, a normalized collision energy of 35% and at a 30,000 mass resolving power (with TurboTMT mode) to resolve the low mass TMT reporter mass. Dynamic exclusion was used for 45 s.

**Peptide identification.** Initial data analysis of raw data generated during this study was carried out using Proteome Discoverer (v 2.1; ThermoFisher Scientific) or MaxQuant (v 1.6.3.4) against the Swissprot *Mus musculus* database (downloaded 04/07/2016; containing 16,795 entries). Searches were conducted with 20 ppm mass tolerance for MS, and 0.2 Da for MS/MS, with one missed cleavage allowed and match between runs enabled. Variable modifications included methionine oxidation, N-terminal protein acetylation, N-terminal methionine cleavage and SILAC-Lys6, Arg10, while the carbamidomethylcysteine modification was fixed. The false discovery rate maximum was set to 0.005% at the peptide identification level (actual was 0.005 for each replicate) and 1% at the protein identification level. All other parameters were left as default.

**Ratio correction and scaling.** Further analysis was performed with custom Python scripts. The logic was as follows. First, the common contaminant protein keratin was removed, leaving 23326 unique peptides associated with 2977 unique proteins. Quantified proteins were considered as those identified by at least two unique peptides, one of which contained a cysteine residue. Although this would allow a protein to be quantified by only a single non-cysteine-containing peptide, the vast majority (89%) of proteins for which a cysteine peptide was identified were associated with more than one non-cysteine peptide. In addition, there was no significant difference in the median non-cysteine abundance ratio between single-non-cysteine-peptide and multi-non-cysteine-peptide proteins. The average peptide abundance ratio (heavy-labeled/light-labeled peptide abundance) for the non-cysteine-containing peptides was calculated for each protein at each urea concentration. The mean non-cysteine abundance ratio was then used to correct the corresponding cysteine-containing peptide(s) for any change in overall protein abundance caused by the treatment, yielding the corrected cysteine ratio. The corrected cysteine ratio was normalized to the native sample (0 M Urea), such that no change resulting from urea denaturation would yield a ratio of 1.

The resultant data was then scaled with a $p$ value weighted correction. This correction weights the mean corrected cysteine ratio of biological replicates ($n = 3$) according to the relative confidence with which it deviates from the expected value (in this case, 1) as per:

$$R = \frac{1 - m}{-20^p} \qquad (2)$$

where $m$ corresponds to the mean of the corrected cysteine ratios and $p$ corresponds to the $p$ value derived from a one-sample $t$-test of the corrected cysteine ratios against the expected value at a single denaturant concentration. To improve confidence in the trends of the corrected peptide ratios across urea concentrations, the data were subsequently smoothed with Locally Estimated Scatterplot Smoothing (LOESS). The resultant curves for the peptide ratios across urea concentrations were clustered by using fuzzy-c means, where the optimal number of clusters was first estimated using the kneed package before manual inspection of ±2 centroids to achieve minimal redundancy in clustered patterns. For all subsequent bioinformatic analyses, peptides were then assigned to clusters with the highest membership score.

**Peptide and protein properties.** The curves for the peptide ratios across urea concentrations were fitted with a sigmoidal two-state unfolding model as per Eq. (3):

$$y = b + \frac{a - b}{1 + e^{\frac{m \times (C_m - [d])}{T \times G}}} \qquad (3)$$

where $a$ and $b$ correspond to the top and bottom plateaus respectively, $C_m$ corresponds to the denaturant concentration at which both the folded and unfolded states are equally populated at equilibrium (assuming two-state protein folding), $[d]$ corresponds to denaturant concentration, and $m$ corresponds to the slope. $T$ and $G$

correspond to the temperature (298.15 K) and gas constants, respectively. Fits were filtered according to the following criteria: (1) $R^2 > 0.75$, (2) absolute value of $a$ and $b$ less than 10, (3) fitted $C_m$ within the range of denaturant concentrations tested, (4) relative error in $C_m$ less than 0.5, and (5) value at 0 M urea in greater than at 6 M urea. The fitted $C_m$ values were then compared against the published datasets as described above.

Physicochemical properties for individual cysteine residues, peptides and proteins of interest were compiled from various databases and extraction/prediction platforms, including UniProt [https://www.uniprot.org/], PFAM [https://pfam.xfam.org/], Protein Data Bank [https://www.ebi.ac.uk/pdbe/], DSSP [https://swift.cmbi.umcn.nl/gv/dssp/][47,48], IUPred2A [https://iupred2a.elte.hu/][49], STRINGdb [https://string-db.org/api/][50], PantherGOSlim [http://pantherdb.org], and iFeature [https://ifeature.erc.monash.edu/][51].

**HSP70 client denaturation assay.** Pig heart L-malate dehydrogenase (MDH2; Roche, catalog # 10127256001) and recombinant human HSPA8 were prepared in HEPES buffer (50 mM HEPES, pH 7.5, 50 mM KCl, 5 mM MgCl₂, 2 mM DTT) to a final concentration of 5 μM and 2 μM, respectively, in the presence or absence of recombinant human DNAJB1 (1 μM). In the case of heat-denatured samples, MDH2 aliquots were heated to 42 °C for 10 min then returned to 37 °C in a thermocycler (BioRad), while native samples were maintained at 37 °C. MDH2 was then combined with the remaining reaction components in the absence or presence of 2 mM ATP (New England Biosciences, catalog # P0756S), then incubated at 37 °C for 30 min in a heating block. In the case of urea-denatured samples, the entire reaction was prepared in the absence or presence of 6 M urea then equilibrated for 4 h at 25 °C in a heating block. Samples were then labeled with 100 μM TPE-MI for 15 min at 25 °C before being diluted into 1 ml ice cold acetone and incubated at –20 °C overnight. Samples, including a pooled control sample, were prepared for mass spectrometry using the Preomics iST-NHS (Preomics, catalog # P.O.00026) and TMT 11-plex labeling (ThermoFisher, catalog # A37725) kits according to the manufacturer's protocol. The pooled channel was added to each biological replicate to support efficient normalization between replicates. Resultant peptides were analyzed using a TMT based MS methodology as described above. The collected spectra were searched against a custom database containing sequences for HSPA8 (P11142), DNAJB1 (P25685) and MDH2 (P00346) sequences downloaded from UniProt. The search was conducted as above, with the following alterations: the MS2 reporter ion was set to TMT 11-plex, isotopic distribution correction applied according to the product data sheet and the fixed carbamidomethylcysteine was replaced with the Preomics alkylation (+113.084 Da).

Filtering and further analysis of the dataset was then carried out with custom Python scripts. The logic was as follows. Raw intensities for peptides with no missed cleavages were scaled according to the molar contribution of the corresponding protein to each reaction. The mean peptide abundance ratio for non-cysteine peptides in each protein that were quantified across all channels containing that protein, was then calculated. The non-cysteine intensity was used to correct the corresponding cysteine-containing peptide(s) for any change in overall protein abundance, resulting in the corrected cysteine ratio. In the case of HSPA8 peptides, the corrected cysteine ratio was then normalized to the native HSPA8 sample, and finally the change in corrected cysteine ratio is reported as the mean of two technical replicates.

**HSP70 inhibition assay.** Neuro-2a cells were seeded and grown for at least 18 h and described above. Culture media was then removed and replaced with fresh media containing VER155008 (Sigma) to a final concentration of 20 μM, or an equivalent amount of DMSO in the case of the vehicle control. Cells were then incubated for a further 18 h before harvesting, lysis, denaturation and TPE-MI labeling as described above. Samples, including a pooled control sample, were prepared for mass spectrometry using the Preomics iST-NHS (Preomics, catalog # P.O.00026) and TMT 11-plex labeling (ThermoFisher, catalog # A37725) kits according to the manufacturer's protocol. A reference channel consisting of a pooled aliquot of all samples was added to each biological replicate to support efficient normalization between replicates. Resultant peptides were analyzed and quantified using the TMT based MS methodology as described above.

Filtering and further analysis of the dataset was then carried out with custom Python scripts. The logic was as follows. First, the common contaminant protein keratin was removed and quantified proteins were considered as those identified by at least two unique peptides, one of which contained a cysteine residue, in at least two out of three biological replicates. The abundance ratio (VER155008/Control) was then calculated for each peptide at each urea concentration. The mean per-protein non-cysteine abundance ratio at each urea concentration was then used to correct the corresponding cysteine-containing peptide(s) for any change in the overall abundance of individual proteins caused by the treatment, yielding the corrected ratio. The resultant data was then scaled using $p$ value weighted correction and $\log_2$ transformed. The standard deviation of all non-cysteine corrected ratios for a given protein across all denaturation concentrations was then calculated as the per-protein threshold. Finally, cysteine-containing peptides quantified in at least 7 denaturant concentrations were collected, and those for which at least one datapoint exceeded the non-cysteine threshold were considered potential hit proteins.

**Statistical analysis**. Statistical analyses were performed either using the scipy module in python[52] or using GraphPad Prism (v 8.4.3). The exact *p* values, raw values and statistical details are provided in the Source Data and Supplementary Data 1.

**Reporting summary**. Further information on research design is available in the Nature Research Reporting Summary linked to this article.

## Data availability

The data that support this study are available from the corresponding author upon reasonable request. Details for the published datasets examined in this study are summarized in Supplementary Table 1. The mass spectrometry proteomics data generated in this study have been deposited in the ProteomeXchange Consortium via the PRIDE[53] partner repository database under accession codes PXD022587 (HSP70 client denaturation), PXD022640 (cell lysate) and PXD030567 (HSP70 inhibition). Protein structures presented in Fig. 4, Fig. 6 and Supplementary Fig. 1 are available via the PDB: 1CJ5, 4H5R, 3AGY, 4KBQ, 3AGZ, 1HDJ, 6NRC. All other data used in this study are available from Zenodo via the https://doi.org/10.5281/zenodo.4280620 [https://doi.org/10.5281/zenodo.4280620]. In addition, select summary datasets are provided in Supplementary Data 2, Supplementary Data 3, Supplementary Data 4 and Source Data files. Source data are provided with this paper.

## Code availability

Custom python analysis scripts used in this study are available from Zenodo via the https://doi.org/10.5281/zenodo.4287766 [https://doi.org/10.5281/zenodo.4287766].

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

## Acknowledgements

We thank Professor Pierre Goloubinoff from University of Lausanne for provision of unfolding data of DNAJB1 (in Supplementary Fig. 6C). We thank Professors Paul Gooley and Heath Ecroyd for helpful discussions and careful reading of the manuscript. We also thank Dr. Yuning Hong (La Trobe University) for providing TPE-MI, and the Bio21 Melbourne Mass Spectrometry and Proteomics facility. This work was funded by grants National Health and Medical Research Council APP1161803 (D.M.H.) and Australian Research Council DP170103093 (D.M.H. and G.E.R.).

## Author contributions

D.C., N.B.N., G.E.R., and D.M.H. designed the research; D.C. and C.A. performed the research and analyzed the data; and D.C. and D.M.H. wrote the manuscript.

## Competing interests

The authors declare no competing interests.
