## [Peer Review File · Nature Communications]

REVIEWER COMMENTS

Reviewer #1 (Remarks to the Author):

This manuscript reports to have uncovered hidden information about protein conformational changes, ligand binding and protein network organization embedded in proteome wide studies of protein folding stability using chemical denaturant-based residue labeling techniques, like SPROX and the author's cysteine labelling approach. As part of this work the authors initially performed a comparative analysis of published protein folding stability data generated using several different MS-based strategies previously developed for collecting such data on the proteomic scale. The results of this comparative analysis revealed consistencies across different studies when the same techniques were used, but not when different techniques were used. For example, poor correlations were observed between the SPROX and thermal profiling methods. While interesting, these results are not surprising as the techniques report on very different biophysical properties of proteins. More interesting would be a discussion/investigation of why some proteins are consistent and others are not. Another component of this manuscript describes the use of a cysteine labelling reagent, TPE-MI, to generate chemical denaturation curves first on a model protein, beta-lactoglobulin, and then on a proteome wide scale using the proteins in a Neuro2a cell lysate. This strategy is conceptually identical to SPROX, and therefore it is not surprising that the proteomic data generated in the current work using TPE-MI was very similar to that generated in SPROX. Indeed, many of the authors' results and conclusions from their TPE-MI clustering results are very similar to those already reported in reference 15 where it was found that ~20% of the met probes were consistent with two-state folding, ~25% were consistent with multi-state folding reactions, and the remaining were solvent exposed. The new "discovery" in the current work really centers around the authors "demonstration" that a substantial component of the changes seen in residue labelling datasets are better explained by changes in protein conformation and ligand interactions that unfolding. The authors' work on the reconstituted HSPA8 system certainly a convincing example of this. However, it is not clear that this is necessarily what is going on in the lysate experiment or that this is a general explanation for the observed proteins with increased cysteine protection observed at increasing denaturant. Along these lines, are some concerns about the model protein data and that these concerns may impact the authors' interpretation of the proteomic data generated (see below).

Major Concerns:

1) The TPE unfolding data in Fig. S1 is difficult to understand. In particular it is not clear how the data in Figure S1B was used to generate the data in Figure S1C. Specifically, how was fraction unfolded calculated? This is not explained in sufficient detail, and it raises questions as to whether or not it was done in the "conventional manner". It would be useful to show the "raw data for both the TPE and TRP curves (i.e., the "signal" vs. [Urea]) As presented, the TPE unfolding data in Fig S1C seems to suggest that the protein completely unfolds at 3 M urea, and then somehow partially refolds

between 4-6 M urea? Not sure this has ever been observed for this model protein. And this seems to be different from what is observed using the Trp probe. And moreover, based on the data in Figure S1, it is hard to understand the data in Figure S1D....The two denaturation curves simply do not overlap, so there is no way they could have the same C_m as indicated in Fig 1SD. This raises questions as to whether or not the TPE experiment is "valid." Perhaps, the TPE data in Fig S1C can be explained by more and more disulfide bond formation in the 4-6 M range and then an apparent "protection" from TPE modification? Given the reaction rate associated with TPE modification it is not surprising that the midpoint of the TPE curve is to the left of the TRP curve.

2) Given the data on the model protein in Fig S1 and the issues brought up above in (1), maybe it is not surprising that some proteins show increased protection at higher denaturant concentrations in the proteome-wide TPE experiment. And based on the model (purified) protein result, maybe there are explanations that do not involve interactions with other proteins. Indeed, there could certainly be more explanation(s) than the hypothesis that the authors' propose. For example, what the authors note in line 236 that is happening in Figure 4A could be the same thing that is going on with the purified model protein (i.e., apparent protection at high denaturant concentration).

3) The in vitro results performed in the presence and absence of ATP are convincing. The authors conclusions would be stronger if they lysate experiment were also performed in the presence and absence of ATP, and the results mirrored those obtained in vitro.

4)The authors conclusions would be stronger if the same results with the reconstituted HSPA8 system and the lysate were obtained with SPROX. This would certainly negate many of the above concerns associated with the thiol reaction.

Minor Issues:

1) Line 45: Given the nature of references 6-13...the appropriate SPROX reference should be West et al in PNAS 2010.

2) The authors never mention how many data points they collected in their proteome-wide TPE experiment. This should be done in line 85 of the SI.

3)Not clear why the model protein study used 50 uM TPE-MI reagent and the proteomic study used 100 uM for 30 min. Why the increase to 100 uM and what is significant about 30 min?

4)Line 143 of the SI: Seems odd to include proteins with only 2 unique peptides (one cys-containing and one not). This means normalizing factor is based on only one peptide. Seems like this might introduce unnecessary error. Why not use proteins with at least 2 or even 3 non-cys peptides?

5) Line 195-198: It is not clear what is meant by "coordinated changes in cysteine reactivity" ?

6)Line 207-212: It is not clear that these conclusions are supported by the data.

7)Line 212: In the parallel analysis of the methionine oxidation data, it is not clear how wt and oxidized peptides were treated (i.e., did the analysis use both, pairs, or all peptides). Seems like to be "parallel", only wild-type met peptides should be considered.

In summary this is a manuscript that does a nice job of documenting important comparisons between different proteomic methods for evaluating protein folding stability. These results are likely to be of interest to those directly using the described techniques, but the results are unlikely to appeal to the broad readership of Nat. Comm. The new knowledge the manuscript attempts to present (ie, the demonstration that a substantial component of the changes seen in residue labelling datasets are better explained by changes in protein conformation and ligand interactions that unfolding) is not sufficiently substantiated in the manuscript. Additional experimental results are needed to substantiate such conclusions.

Reviewer #2 (Remarks to the Author):

In this manuscript titled "Hidden information on protein function in censuses of proteome foldedness" by Cox et al, the authors describe a framework to understand how proteins unfold and interact with ligands/cofactors in complex cellular environments. The main findings of the paper are that proteome-wide unfolding/stability studies largely do not correlate, in particular comparing experiments using different means of denaturation (i.e. thermal, GuHCl and urea) deriving T_m or C_m values for individual proteins in the proteome. Leveraging a specific thiol probe (TPE-MI) the authors demonstrate unfolding of a model substrate (as compared to trp fluorescence). The authors then apply this reagent to proteome-wide experiments and use urea to monitor changes in thiol modification frequency with TPE-MI. The derived stability values are consistent with previously published data using a similar approach. A simple statistical analysis of the changes in cysteine peptide abundance reveals a breadth of information which the authors deconvolute into distinct processes: 1) unfolding, 2) unfolding/binding and 3) mixed interactions. The authors then perform in vitro experiments using a chaperone/co-chaperone/substrate to validate their observations from proteome-wide experiments. The observations that binding events may be coupled to unfolding events can explain the data is intriguing but more experiments are required to test the hypothesis. The study is well-written, important and of wide interest to the protein folding field. Outside of several concerns that should be addressed with additional experiments and revisions, the study may be suitable for publication in Nature Communications.

Major points

1. The authors find that their proteome-wide unfolding data may be explained by not only unfolding events but also by binding events between denatured substrates and chaperones under denaturing conditions. While I find this idea intriguing it would be really important to test this hypothesis in more detail using a model system (see below) to convince the reader that this indeed can happen and that there are no alternate explanations for the observation. Naturally, I would think that under denaturing conditions chaperones can also unfold but even if they remain folded, binding events under chemical denaturation conditions would also likely be severely attenuated and thus unlikely to be abundant.

2. The in vitro validation of the HspA8 and DnaJB1 experiments are a very important part of this manuscript because they are the direct validation of the proposed hypothesis. A more careful biochemical analysis of the in vitro system proposed would be central to testing the chaperone/co-chaperone/substrate binding observations. At the proposed denaturing conditions, how folded are the different components? Does mixing them change their stability? Does ATP change the stability of the components? This may demonstrate that the chaperone is folded sufficiently to still be able to recognize a substrate under mildly denaturing conditions. Of course, an important caveat between the lysate and in vitro experiments is the complex milieu present in lysates and what contribution this has on stability of proteins. Finally, in addition to foldedness/activity of the components, it would also be important to demonstrate that these binding events can be detected and modulated with ATP as proposed. Perhaps the TPE-MI experiments could be coupled to crosslinking experiments to show the relationship between cysteine exposure and binding events.

3. It is very interesting that the eukaryotic chaperonin TRiC/CCT is identified in the unfolding/binding proteome screen. It might be worth expanding the discussion of this data in the results given there is structural information of the complex and it is known how it interacts with canonical substrates such as tubulin or actin. Surely there must be consistency between the observed clusters in TRiC/CCT with other chaperones including HspA8/DnaJB1/substrate.

4. It is not clear how the peptides are normalized to quantify cysteine reactivity. This is important as it is difficult to compare intensities in mass spectrometry across different peptide sequences (Page4, Line105-107). In the normalization, what is the control and how confident are the authors that this is a suitable control – the observed changes in cysteine reactivity are crucially dependent on this signal.

5. In terms of proteomic level analysis, is the protein number in Figure 3C meaningful especially since the authors try to show their labeling method can appropriately explain the two-state folding on large scale.

6. For denaturing, the author use Fig S1B as control with up to 3.5M Urea. However, the proteomic data use up to 6M. If the controls shows no data above 3.5M Urea, I am not sure the Fig2B and Fig4A is reliable when the curve change only in higher Urea.

Minor points

1. For Fig. 4A, the authors state in the text that the peptides with cysteines in DnaJB1 presented “overall decreased reactivity” through increased urea gradient but looking at the figure one does not observe the data to support the statement. If anything, a slight trend of “more reactive” could be concluded instead.

2. On line 227, the authors state that Fig. 4A “showed 3 cysteine peptides in different clusters” which can be understood that all 3 peptides are in 3 different clusters, which is not the case – 3 cysteine peptides were observed in only 2 different clusters.

3. Fig. 4B doesn't seem very intuitive and it is not explained either in the text or the figure legend. Please clarify the description of this panel.

4. I believe there is a typo on line 262 where the authors describe Fig. 4E. There is no such figure, this is likely Fig. 4C.

Reviewer #3 (Remarks to the Author):

In the manuscript ‘Hidden information on protein function in censuses of proteome foldedness’ Hatters and colleagues start off by finding little correlation in proteome-wide stability measurements when determined with different methods. They find that this is at least in part due to significant contribution of factors such as ligand binding and protein interactions etc that influence stability measurements in various ways that cannot be easily explained with simple unimodal

denaturation. In particular methods assessing amino acid side chain reactivity are identified as sensitive to conformation changes and ligand binding. They further find that a surprisingly high number of cysteine or methionine residues in mammalian proteomes displayed reduced reactivity upon denaturation. This paradoxical behavior was enriched in proteins with functions relating to unfolded protein stress and a more detailed analysis is provided for the chaperone HSP8 where the effects of ATP on side chain reactivities were studied.

In this reviewers' opinion, the authors correctly state that various factors influence side chain reactivity in their cysteine alkylation approach using a large hydrophobic maleimide and similar considerations will plausibly apply for other labeling approaches. However, the finding that the various methods assessing protein stability and conformation in complex biological matrices assess properties that are influenced by various factors other than simple unfolding is well established. That said there are several interesting features found for their specific labeling approach that related to detecting subtle conformational changes that are at the moment somewhat underdeveloped in the manuscript and might warrant deeper analysis and further investigation when revising the manuscript for publication in this journal or another one.

Main points:

- The authors start off stating that proteome wide measurements of thermal stability, proteolysis resistance and reactivity of amino acid side chains provide a global census of protein folding with chemical denaturation midpoints and thermal melting midpoints being interpreted as correlates of ΔG . It should be noted though that whilst C_m , T_m and other measurements might have practical value for characterizing protein stability in a defined state in a complex matrix, main focus in the scientific literature is on measuring the delta of these values when comparing two sets of experimental conditions, e.g. cellular treatments or addition of ligands. For example, the differences when comparing protein T_m s in cell extracts vs live cells and even when comparing different protein concentrations have been widely discussed in the scientific literature. Similarly, protease resistance and proteolytic susceptibility of individual sequence stretches is not predominantly interpreted as an absolute value but as an experiment specific measure of protein fold that might alter upon treatments. For the majority of these methods detailed investigations have been reported as to how ligand binding and post translational modifications might alter these measurements with protein global (e.g. doi: 10.1016/j.cell.2018.03.053, <https://pubmed.ncbi.nlm.nih.gov/30858367/>) or (sub-)domain (e.g. [https://www.cell.com/cell/pdf/S0092-8674\(17\)31448-4.pdf](https://www.cell.com/cell/pdf/S0092-8674(17)31448-4.pdf)) resolution. Such detailed investigations may have been missing for the specific global protein labeling field in particular with the here described cysteine labeling approach but the authors should take greater care when generalizing their findings in the context of what is known for other methods. In particular the Lip-MS approach pioneered by the Picotti lab offers similar opportunities to detect locally restricted conformational changes and has been demonstrated to allow localizing ligand binding. This work is currently only insufficiently appreciated in introduction and discussion and it should be clarified that many of the conceptual findings made for side chain reactivity in this paper have been reported on global scale with the LIP-MS approach, previously. Possible advantages of the side chain labeling approach might also become more apparent upon a more detailed comparison of the data with the proteolysis approach.

- Figure 1: Thermal profiling might be over-represented. In general, for intra-method comparisons marked differences in protocol should be marked: experiments performed in lysate vs cells, also there are strong differences in published lysate protocols. To what extent could such differences also have affected the very poor correlation of residue labeling experiments? As residue labeling and proteolysis both have the potential to obtain site level resolution, the level of concordance on a residue level would be very informative. For the rescaled measures of protein stability such as T_m and C_m the authors would need to prove that the distribution of data between 0 and 1 will be comparable for the different approaches

- The TME_MI reagent was previously mainly used to detect the foldedness of the total proteome but less so to derive mechanistic insights in unfolding events with protein and sub-protein resolution as applied here. This comes with an increased need to provide evidence that the label does not lead to unfolding and localized structural rearrangements by itself, e.g. due to the excessive hydrophobicity of the modification compared to the free cys. Validation with different reagents or independent methods (such as proteolysis) would be needed to back the claims made for individual proteins (e.g. HSP8).

Likewise, comparisons to methionine oxidation studies need to be taken with a pinch of salt as oxidation can have profound and long ranging structural impact that is difficult to distinguish from direct local unfolding in differential experiments (e.g.

<https://www.sciencedirect.com/science/article/pii/S092544390900297X>)

- At high concentrations of denaturants difficult to control parameters include the denaturing effect of covalent ligands such as the TPE-MI reagent and increasing oxidation effects that may make individual residues appear less reactive to other reactants. Hence, it is important to quantify the probe reacted and the non-probe reacted fraction of cysteine residues of interest to keep control of the total balance of the monitored cys residues. This is particularly important when trying to explain complex reactivity patterns such as the ones for HSP8 NBD. Unfortunately, the described approach only uses the changes in abundances of peptides with unreacted cysteins for quantification only (p4). - Is this because of the TPE-MI label causes a strong hydrophobic shift and unfavorable fragmentation? Please discuss.

- I have strong reservations with the interpretation of clusters in Figure 2B: Cluster 4 looks like cluster 1: unchanged reactivity with just a bit more variable data and potentially a slight increase around 1 M. Similarly, clusters 2 and three are essentially the same just that 3 contains more low quality data with a similar but opposite effect around 1 M urea. For 1 and 4 no wonder that the fits are rather poor as there are essentially no changes, for clusters 2 and 3 only goodness of fit distribution is somewhat different. For clusters 1 and 4 concordance of no change would be more informative than the correlation. The dataset shown in Figure 3E that was derived from a different publication seems to be of better data quality and Cluster 4 indeed indicates reduced reactivity of methionine residues. It should be noted though that methionine labeling is even more susceptible to oxidation artifacts than Cys labeling.

- Figure 3 not particularly informative. D is kind of self-explaining as larger proteins tend to have more domains and can interact with more proteins than shorter ones. The entire figure might be better placed in supplemental.

- HSP8 is a very complex protein to study: due to its roles in the cytosol but also in the lysosome as main carrier for chaperone mediated autophagy. Depending on experimental protocols access to the different subcellular compartments might be different at different denaturant conditions. This affects argumentation provided in line 253 ff for example. Changes observed for individual Cys residues are very small and even for the just significant ones the differences of the mean are much below 0.1. For Cys17 spread is just bigger so one cannot conclude that the data for 17 are significantly different to those for 574. Whilst the conclusions may seem plausible, they lack validation.

Minor point

- Figure 4 could do with a better structural model of HSPA8. It might be worth including the one from Figure S6. Suggest depicting the various nucleotide binding motifs as well. C:

Reviewer #4 (Remarks to the Author):

This manuscript describes an approach to identify information on protein conformation and ligand interactions from denaturation data. The study determines that datasets for published denaturation studies do not only provide information on protein folding but also contain information regarding other structural aspects. Further, the authors utilized thiol reactivity labeling to further underscore that these types of studies provide more information than just on protein folding. In particular, this study demonstrated that cysteine peptide reactivity profiles could be defined by four distinct clusters, two of which closely resemble a two-state unfolding process. A decrease in cysteine reactivity in peptides in cluster 4 show other mechanisms of structural changes occurring such as interactions or conformational changes that are revealed in the data. The experimental design is well thought out and the proper controls have been analyzed. However, there are some significant changes in the manuscript that need to be made prior to publication.

-The Methods sections states that 12 studies were deemed "suitable" for the pairwise analysis. What was the criteria used to make these 12 "suitable?"

-On page 6 line 161, it states clusters 2 and 3 were likely to contain hydrophobic and solvent-buried residues. This is the case for cluster 2 in Figure 3B but not so for cluster 3 when it comes to buried residues.

-The authors should provide some sentences to explain PFAM where it is first mentioned on page 6 line 169. This term may not be commonly known.

-Page 7 line 186 refers to Figure S3E, I believe this should be Figure S2E

-Page 8 line 213 refers to Figure 3G, I believe this should be 3H

-Page 8 line 238 describes residues Cys267 and 269 as having overall decreased reactivity. It appears from figure 4A that the reactivity is actually increased. The authors need to explain this.

Reviewer #1 (Remarks to the Author):

This manuscript reports to have uncovered hidden information about protein conformational changes, ligand binding and protein network organization embedded in proteome wide studies of protein folding stability using chemical denaturant-based residue labeling techniques, like SPROX and the author's cysteine labelling approach. As part of this work the authors initially performed a comparative analysis of published protein folding stability data generated using several different MS-based strategies previously developed for collecting such data on the proteomic scale. The results of this comparative analysis revealed consistencies across different studies when the same techniques were used, but not when different techniques were used. For example, poor correlations were observed between the SPROX and thermal profiling methods. While interesting, these results are not surprising as the techniques report on very different biophysical properties of proteins. More interesting would be a discussion/investigation of why some proteins are consistent and others are not. Another component of this manuscript describes the use of a cysteine labelling reagent, TPE-MI, to generate chemical denaturation curves first on a model protein, beta-lactoglobulin, and then on a proteome wide scale using the proteins in a Neuro2a cell lysate. This strategy is conceptually identical to SPROX, and therefore it is not surprising that the proteomic data generated in the current work using TPE-MI was very similar to that generated in SPROX. Indeed, many of the authors' results and conclusions from their TPE-MI clustering results are very similar to those already reported in reference 15 where it was found that ~20% of the met probes were consistent with two-state folding, ~25% were consistent with multi-state folding reactions, and the remaining were solvent exposed. The new "discovery" in the current work really centers around the authors "demonstration" that a substantial component of the changes seen in residue labelling datasets are better explained by changes in protein conformation and ligand interactions that unfolding. The authors' work on the reconstituted HSPA8 system certainly a convincing example of this. However, it is not clear that this is necessarily what is going on in the lysate experiment or that this is a general explanation for the observed proteins with increased cysteine protection observed at increasing denaturant. Along these lines, are some concerns about the model protein data and that these concerns may impact the authors' interpretation of the proteomic data generated (see below).

Major Concerns:

1. The TPE unfolding data in Fig. S1 is difficult to understand. In particular it is not clear how the data in Figure S1B was used to generate the data in Figure S1C. Specifically, how was fraction unfolded calculated? This is not explained in sufficient detail, and it raises questions as to whether or not it was done in the "conventional manner". It would be useful to show the "raw data for both the TPE and TRP curves (i.e., the "signal" vs. [Urea]) As presented, the TPE unfolding data in Fig S1C seems to suggest that the protein completely unfolds at 3 M urea, and then somehow partially refolds between 4-6 M urea? Not sure this has ever been observed for this model protein. And this seems to be different from what is observed using the Trp probe. And moreover, based on the data in Figure S1, it is hard to understand the data in Figure S1D....The two denaturation curves simply do not overlap, so there is no way they could have the same C_m as indicated in Fig 1SD. This raises questions as to whether or not the TPE experiment is "valid." Perhaps, the TPE data in Fig S1C can be explained by more and more disulfide bond formation in the 4-6 M range and then an apparent "protection" from TPE modification? Given the reaction rate associated with TPE modification it is not surprising that the midpoint of the TPE curve is to the left of the TRP curve.

The reviewer has drawn to our attention the need to better explain this supplementary figure and the analysis that was done. We stress that the data has been analysed by conventional approaches and does not show any anomalous features. We have also shown the raw data. It is indeed quite normal for experimental measures of foldedness to show an underlying linear relationship with denaturant that is unrelated to the actual process of folding, and it is normal for this to be taken into consideration in the fitting of the data in the

way that we have done it (Santoro et al., 1988 Biochemistry). Critical to this point is that the protein is not becoming more folded between 4-6 M as the changes that are happening relate to the underlying relationships described above. We have made a number of changes to the manuscript to clarify the details and show why we concluded that the midpoint of denaturation was not different between the TPE data and the Trp data.

First, we included additional detail in the methods (Supplementary Information), to clarify the derivation and normalisation of the rate of reaction between TPE-MI and β -lactoglobulin across various urea concentrations. The resultant datapoints are shown in panel C, compared directly to the normalised tryptophan intensity values.

We have also added additional detail to the methods pertaining to the procedure of fitting of the data to include calculation of two linear relationships for the pre- and post-unfolding parts of the unfolding curve. The pre- and post-transition states are incorporated as parameters of the fitted model, such that any differences between these states for different methods of observing unfolding are accounted for when deriving the transition midpoint (C_m in this case). As stated above, this analysis is absolutely standard protocol for denaturation data. For clarity, we have also included a corrected version of the denaturation fit, in which the contribution of the pre- and post-transition states is removed from the sigmoidal curve, as an inset to Figure S1 C. We thus arrive at the fitted C_m values presented in panel D (and now shown in Figure S1 C inset), which show no significant difference between the TPE-MI and tryptophan-reported unfolding.

2. Given the data on the model protein in Fig S1 and the issues brought up above in (1), maybe it is not surprising that some proteins show increased protection at higher denaturant concentrations in the proteome-wide TPE experiment. And based on the model (purified) protein result, maybe there are explanations that do not involve interactions with other proteins. Indeed, there could certainly be more explanation(s) than the hypothesis that the authors' propose. For example, what the authors note in line 236 that is happening in Figure 4A could be the same thing that is going on with the purified model protein (i.e., apparent protection at high denaturant concentration).

As described in response 1 above, the data were fitted to a standard two-state model of denaturation. The same model fitting procedure was applied to the lysate denaturation curves to accommodate systematic linear changes to the signal in the pre and post transition folding. Thus, even when accounting for pre- and post-transition effects resulting from TPE-MI dye, the response measured for peptides such as those highlighted in Figure 4 is not consistent with a two-state denaturation model.

3. The in vitro results performed in the presence and absence of ATP are convincing. The authors conclusions would be stronger if they lysate experiment were also performed in the presence and absence of ATP, and the results mirrored those obtained in vitro.

The results of such an experiment will not be straightforward to interpret. While we can attribute binding in the case of chaperones such as HSP70 and HSP40 to the absence of ATP and presence of unfolded client in the simple reconstituted system, there are a spectrum of protein-protein or protein-ligand interactions that may otherwise contribute to the protection observed within the lysate experiment. As such, we believe it is not a reasonable expectation that the addition of ATP (which we assume to be limiting in the case of the lysate experiment over 4 h equilibration in urea) would resolve these interactions within the complex cellular milieu observed as different protection phenotypes.

As an alternative, we examined changes in the cys-reactivity in lysates of cells treated with the small molecule VER155008, that is a specific competitive inhibitor to the ATP binding pocket of HSP70 family proteins (new data included as Fig 5 in the revised manuscript).

These data address the concern raised by the reviewer and support the hypothesis that the protection we observe in HSPA8 is associated with engaging in interaction with known binding partners.

4. The authors conclusions would be stronger if the same results with the reconstituted HSPA8 system and the lysate were obtained with SPROX. This would certainly negate many of the above concerns associated with the thiol reaction.

The concerns raised by the reviewer about the thiol reactivity are addressed above in points 1 and 2. However, we did consider SPROX as an approach to validate the findings. Reviewing the sequence and structure of HSPA8 reveals there are no methionine residues in the quantified C-terminal cys peptides, and the location of the nearest-neighbouring methionine residues places them outside the α -helices containing the measured cys-peptides. For this reason, we anticipate that SPROX in the context of this particular recombinant assay would provide complementary, but not necessarily confirmatory, information. Indeed, these residues may very well experience different solvent environments to the quantified cysteine-containing peptides and thus interpreting comparative SPROX data at this fine scale may not be straightforward.

We would note, however, that at the bulk proteome level where many more peptides were observed to contain *both* reactive cysteine and methionine residues, the same-species comparison of SPROX data (Roberts et al., 2016, J Proteome Res.) with the TPE-MI data presented here was one of only two datasets that were significantly correlated across all proteins. This suggests that our method of residue labelling provides, as a whole, consistent information regarding protein foldedness as that derived from SPROX studies. The reviewer acknowledged this in their opening paragraph.

Minor Issues:

1. Line 45: Given the nature of references 6-13...the appropriate SPROX reference should be West et al in PNAS 2010.

This reference has been included as appropriate in line 45.

2. The authors never mention how many data points they collected in their proteome-wide TPE experiment. This should be done in line 85 of the SI.

This information has been included in the Supplementary Information (pg 6) as requested, as well as clarification of the biological n in the associated legend (Fig. 2).

3. Not clear why the model protein study used 50 μ M TPE-MI reagent and the proteomic study used 100 μ M for 30 min. Why the increase to 100 μ M and what is significant about 30 min?

TPE-MI concentration and incubation times were modelled from a previous study (Chen et al., 2017, Nat Comms). The only exception was our decision to lower the concentration for labelling of cell lysate to 100 μ M (compared to staining of intact cells at 200 μ M in the previous case), given here we were operating in cell lysate and not reliant on the ability of TPE-MI to cross the cell membrane. Additional detail describing this has been included in the methods (Supplementary material pg 3, 4).

4. Line 143 of the SI: Seems odd to include proteins with only 2 unique peptides (one cys-containing and one not). This means normalizing factor is based on only one peptide. Seems like this might introduce unnecessary error. Why not use proteins with at least 2 or even 3 non-cys peptides?

The vast majority (89%) of proteins for which a cysteine peptide was identified were associated with more than one non-cysteine peptide. In addition, there was no significant difference in the median non-cys abundance ratio between single-non-cys-peptide and multi-non-cys-peptide proteins. These details have been added to the methods (Supplementary Material pg 6).

5. Line 195-198: It is not clear what is meant by "coordinated changes in cysteine reactivity" ?

In essence, this statement is meant to convey the concept that the exposure of one protein from a complex by definition yields exposure of its binding partner. This has been clarified in the corresponding text (pg 7).

6. Line 207-212: It is not clear that these conclusions are supported by the data.

We have reworded this section of the text to indicate that the data (i.e. gene ontology enrichment) supports this mechanism, rather than conclude this mechanism (pg 7).

7. Line 212: In the parallel analysis of the methionine oxidation data, it is not clear how wt and oxidized peptides were treated (i.e., did the analysis use both, pairs, or all peptides). Seems like to be "parallel", only wild-type met peptides should be considered.

As Met-Ox peptides are amenable to quantitation, the Walker dataset provides the opportunity to explore the appearance of the oxidised peptides. In contrast, the TPE-MI measures the loss of unmodified peptides. Thus, while the loss of unmodified methionine peptides would be the most comparable dataset to the presented TPE-MI data from a methodological standpoint, we felt it most useful to perform a comparison to the appearance of modified peptides in the Walker dataset, acknowledging that the two types of data should be inverse to one another. Additional clarification has been included in the text (pg 7).

In summary this is a manuscript that does a nice job of documenting important comparisons between different proteomic methods for evaluating protein folding stability. These results are likely to be of interest to those directly using the described techniques, but the results are unlikely to appeal to the broad readership of Nat. Comm. The new knowledge the manuscript attempts to present (ie, the demonstration that a substantial component of the changes seen in residue labelling datasets are better explained by changes in protein conformation and ligand interactions that unfolding) is not sufficiently substantiated in the manuscript. Additional experimental results are needed to substantiate such conclusions.

Reviewer #2 (Remarks to the Author):

In this manuscript titled "Hidden information on protein function in censuses of proteome foldedness" by Cox et al, the authors describe a framework to understand how proteins unfold and interact with ligands/cofactors in complex cellular environments. The main findings of the paper are that proteome-wide unfolding/stability studies largely do not correlate, in particular comparing experiments using different means of denaturation (i.e. thermal, GuHCl and urea) deriving Tm or Cm values for individual proteins in the proteome. Leveraging a specific thiol probe (TPE-MI) the authors demonstrate unfolding of a model substrate (as compared to trp fluorescence). The authors then apply this reagent to proteome-wide experiments and use urea to monitor changes in thiol modification frequency with TPE-MI. The derived stability values are consistent with previously published data using a similar approach. A simple statistical analysis of the changes in cysteine peptide abundance reveals a breadth of information which the authors deconvolute into distinct processes: 1) unfolding, 2) unfolding/binding and 3) mixed interactions. The authors then perform in vitro experiments using a chaperone/co-chaperone/substrate to validate their observations from proteome-wide experiments. The observations that binding events may be coupled to unfolding events can explain the data is intriguing but more experiments are required to test the hypothesis.

The study is well-written, important and of wide interest to the protein folding field. Outside of several concerns that should be addressed with additional experiments and revisions, the study may be suitable for publication in Nature Communications.

Major points

1. The authors find that their proteome-wide unfolding data may be explained by not only unfolding events but also by binding events between denatured substrates and chaperones under denaturing conditions. While I find this idea intriguing it would be really important to test this hypothesis in more detail using a model system (see below) to convince the reader that this indeed can happen and that there are no alternate explanations for the observation. Naturally, I would think that under denaturing conditions chaperones can also unfold but even if they remain folded, binding events under chemical denaturation conditions would also likely be severely attenuated and thus unlikely to be abundant.

Since our original submission of this work, a paper was published describing the urea denaturation curve of HSPA8 which helps support our findings (Silva et al (2021) *Biochim Biophys Acta Proteins Proteom* 1869, 140719.). To supplement this published result, we have also now included a supplementary figure demonstrating the response of recombinant DNAJB1 to increasing concentrations of urea, measured via intrinsic tryptophan fluorescence (now Fig S6), which also helps explain our findings.

In addition, we have also included the equivalent TPE-MI data obtained in the presence of urea to complement the data originally presented in Figure 4D (now Fig. 4F and S6). These new data were obtained by subjecting the entire reaction mixture to urea denaturation (mimicking the lysate denaturation experiment) rather than just the client as presented previously. While there are some differences to the native chaperone+denatured client data, the results overall are consistent with the changes seen in lysate. A discussion of these results has been added to the text on pg 11.

2. The in vitro validation of the HspA8 and DnaJB1 experiments are a very important part of this manuscript because they are the direct validation of the proposed hypothesis. A more careful biochemical analysis of the in vitro system proposed would be central to testing the chaperone/co-chaperone/substrate binding observations.

At the proposed denaturing conditions, how folded are the different components? Does mixing them change their stability? Does ATP change the stability of the components?

As indicated above for point 1, a recent study has performed a urea denaturation curve of HSPA8 and shows it has a two-domain folding behaviour. One domain, likely the NBD, has a C_m around 1.5 M urea and the other domain a C_m around 3.6 M urea (Silva et al (2021) *Biochim Biophys Acta Proteins Proteom* 1869, 140719.)

In this study ATP was shown to affect the stability of the NBD domain. Collectively these findings are in agreement with our proposed mechanism and we have cited the relevance of this new publication in our revised manuscript.

This may demonstrate that the chaperone is folded sufficiently to still be able to recognize a substrate under mildly denaturing conditions. Of course, an important caveat between the lysate and in vitro experiments is the complex milieu present in lysates and what contribution this has on stability of proteins. Finally, in addition to foldedness/activity of the components, it would also be important to demonstrate that these binding events can be detected and modulated with ATP as proposed. Perhaps the TPE-MI experiments could be coupled to crosslinking experiments to show the relationship between cysteine exposure and binding events.

We performed cross linking experiments using the lysine-cross linker DSSO (disuccinimidyl sulfoxide) in an effort to address this question raised by the reviewer. Unfortunately however, the experiments proved far from straightforward in reality and we were unable to extract information that either supported or refuted the model. In essence, we did not see cross-linked peptides that corresponded to the motifs of interest relevant to our TPE-experiments. This was because we observed very few intramolecular cross-links within the chaperones. This may be because the spatial proximity of lysines were unfavourable to gain the coverage required to make definitive conclusions. However, we feel that the new TPE experiments involving urea denaturation provides an independent experimental strategy to earmark the changes in cysteine reactivity that correspond to substrate binding.

3. It is very interesting that the eukaryotic chaperonin TRiC/CCT is identified in the unfolding/binding proteome screen. It might be worth expanding the discussion of this data in the results given there is structural information of the complex and it is known how it interacts with canonical substrates such as tubulin or actin. Surely there must be consistency between the observed clusters in TRiC/CCT with other chaperones including HspA8/DnaJB1/substrate.

We thank the reviewer for raising this very interesting point. We have now revisited the chaperone peptide data and expanded our analysis of this dataset to include the TRiC machinery, as well as other well-characterised chaperone families such as HSP90s. These results are presented as Fig 6, Supplementary table S2, and are accompanied by a discussion of these findings (pg 12). In essence, we do find additional evidence of protection of key residues or regions among chaperone families in addition to HSPA8, including TRiC.

4. It is not clear how the peptides are normalized to quantify cysteine reactivity. This is important as it is difficult to compare intensities in mass spectrometry across different peptide sequences (Page4, Line105-107). In the normalization, what is the control and how confident are the authors that this is a suitable control – the observed changes in cysteine reactivity are crucially dependent on this signal.

The normalisation method is detailed in the Supplementary Materials method text (as per the journal requirements; pg 6 line 158-164). In essence, quantification of an individual peptide is first reported as a ratio between the light- (minus urea) and heavy-labelled (plus urea) abundance of that peptide. This avoids any confounding issues of comparing the raw abundance across different peptides. This ratio is then corrected for any change in the overall abundance of the parent protein that is not due to cysteine reaction, as measured by the average Heavy/Light ratio of non-cysteine containing peptides for a given protein. Finally, the resultant corrected ratio is normalised to 0 M urea, such that we report the change in cysteine reactivity due to increasing denaturant concentration. We are confident that this method of correction per-protein (i.e. using the non-cys containing peptides) is a suitable internal control that allows us to account for any changes in abundance not related to cysteine reactivity.

5. In terms of proteomic level analysis, is the protein number in Figure 3C meaningful especially since the authors try to show their labeling method can appropriately explain the two-state folding on large scale.

We have added this panel to provide an overview of the distribution of proteins assigned into specific clusters. The information is important to show because it demonstrates that it is only a small fraction of the proteins that can be robustly explained by two-state unfolding.

6. For denaturing, the author use Fig S1B as control with up to 3.5M Urea. However, the proteomic data use up to 6M. If the controls shows no data above 3.5M Urea, I am not sure the Fig2B and Fig4A is reliable when the curve change only in higher Urea.

The supplementary figure (Fig S1) demonstrates the ability of TPE-MI to monitor two-state unfolding in a simple globular protein. However, this data is not intended to act as a control for the denaturation data derived from complex mixtures via proteomics. Instead, the proteomics data use an inbuilt per-protein control (the non-cys peptides; as described in response to point 4 above and on pg 4). It is anticipated that, within the proteome, proteins would exhibit a spectrum of transition points ranging across the higher urea concentrations, as observed in figure 2B cluster 2 and 3.

Minor points

1. For Fig. 4A, the authors state in the text that the peptides with cysteines in DnaJB1 presented “overall decreased reactivity” through increased urea gradient but looking at the figure one does not observe the data to support the statement. If anything, a slight trend of “more reactive” could be concluded instead.

We thank the reviewer for pointing out this inconsistency – this is correct, and the text has been updated accordingly.

2. On line 227, the authors state that Fig. 4A “showed 3 cysteine peptides in different clusters” which can be understood that all 3 peptides are in 3 different clusters, which is not the case – 3 cysteine peptides were observed in only 2 different clusters.

We thank the reviewer for pointing out this ambiguity and have updated the text accordingly.

3. Fig. 4B doesn’t seem very intuitive and it is not explained either in the text or the figure legend. Please clarify the description of this panel.

We have now added additional explanation to the figure legend for this panel as requested.

4. I believe there is a typo on line 262 where the authors describe Fig. 4E. There is no such figure, this is likely Fig. 4C.

We thank the reviewer for their careful review of the manuscript and have updated the figure reference.

Reviewer #3 (Remarks to the Author):

In the manuscript ‘Hidden information on protein function in censuses of proteome foldedness’ Hatters and colleagues start off by finding little correlation in proteome-wide stability measurements when determined with different methods. They find that this is at least in part due to significant contribution of factors such as ligand binding and protein interactions etc that influence stability measurements in various ways that cannot be easily explained with simple unimodal denaturation. In particular methods assessing amino acid side chain reactivity are identified as sensitive to conformation changes and ligand binding. They further find that a surprisingly high number of cysteine or methionine residues in mammalian proteomes displayed reduced reactivity upon denaturation. This paradoxical behavior was enriched in proteins with functions relating to unfolded protein stress and a more detailed analysis is provided for the chaperone HSP8 where the effects of ATP on side chain reactivities were studied.

In this reviewers’ opinion, the authors correctly state that various factors influence side chain reactivity in their cysteine alkylation approach using a large hydrophobic maleimide and similar considerations will plausibly apply for other labeling approaches. However, the finding that the various methods assessing protein stability and conformation in complex biological matrices assess properties that are influenced by various factors other than simple unfolding is well established. That said there are several interesting features found for their specific labeling approach that related to detecting subtle conformational changes that are at the moment somewhat underdeveloped in

the manuscript and might warrant deeper analysis and further investigation when revising the manuscript for publication in this journal or another one.

Main points:

1. The authors start off stating that proteome wide measurements of thermal stability, proteolysis resistance and reactivity of amino acid side chains provide a global census of protein folding with chemical denaturation midpoints and thermal melting midpoints being interpreted as correlates of DG. **It should be noted though that whilst C_m , T_m and other measurements might have practical value for characterizing protein stability in a defined state in a complex matrix, main focus in the scientific literature is on measuring the delta of these values when comparing two sets of experimental conditions, e.g. cellular treatments or addition of ligands.** For example, the differences when comparing protein T_m s in cell extracts vs live cells and even when comparing different protein concentrations have been widely discussed in the scientific literature. Similarly, protease resistance and proteolytic susceptibility of individual sequence stretches is not predominantly interpreted as an absolute value but as an experiment specific measure of protein fold that might alter upon treatments. For the majority of these methods detailed investigations have been reported as to how ligand binding and post translational modifications might alter these measurements with protein global (e.g. doi: 10.1016/j.cell.2018.03.053, <https://pubmed.ncbi.nlm.nih.gov/30858367/>) or (sub-)domain (e.g. [https://www.cell.com/cell/pdf/S0092-8674\(17\)31448-4.pdf](https://www.cell.com/cell/pdf/S0092-8674(17)31448-4.pdf)) resolution. Such detailed investigations may have been missing for the specific global protein labeling field in particular with the here described cysteine labeling approach but the authors should **take greater care when generalizing their findings** in the context of what is known for other methods. In particular the Lip-MS approach pioneered by the Picotti lab offers similar opportunities to detect locally restricted conformational changes and has been demonstrated to allow localizing ligand binding. This work is currently only insufficiently appreciated in introduction and discussion and it should be clarified that many of the conceptual findings made for side chain reactivity in this paper have been reported on global scale with the LIP-MS approach, previously. Possible advantages of the side chain labeling approach might also become more apparent upon a more detailed comparison of the data with the proteolysis approach.

We appreciate the points raised here by the reviewer. We have indeed come to this research problem from the opposite direction (ie protein folding-misfolding) to how many others have been employing stability measurements as proxies to solving problems of ligand binding etc. While it might be apparent to the reviewer that the measures of stability do not clearly report on simple folding stability, we agree there is value in making a clear point on this.

We have therefore revised the abstract, introduction and discussion sections to better acknowledge the points raised by the reviewer including adding the references mentioned by the reviewer. We have in addition retained our key point that providing a pathway to delineate which of these reported measures of stability relate to simple folding stability versus other features different to stability such as ligand binding. A key point of our paper is that it does describe how to delineate these from each other. We feel that these changes significantly improve the clarity of the manuscript and provide a valuable addition to the prior work that has suggested that thermal stability measures can provide a proxy for conformational changes arising in proteomes by providing a workflow on how to detect such changes.

2. Figure 1: Thermal profiling might be over-represented. In general, for intra-method comparisons marked differences in protocol should be marked: experiments performed in

lysate vs cells, also there are strong differences in published lysate protocols. To what extent could such differences also have affected the very poor correlation of residue labeling experiments? As residue labeling and proteolysis both have the potential to obtain site level resolution, the level of concordance on a residue level would be very informative. For the rescaled measures of protein stability such as T_m and C_m the authors would need to prove that the distribution of data between 0 and 1 will be comparable for the different approaches.

The datasets included in the comparison here were chosen by systematic literature search. At the initial stage of this search, we used a set of broad search terms designed to capture any examples of the application of proteomics-based measures of stability. From there, we used specific criteria (experimental parameters such as providing a control whole-proteome stability measure, data availability and species). Any over-representation of the thermal profiling method stems from its over-representation in the available literature. These details are provided as part of the methods on pg 2 of the supplementary information.

An underlying assumption of these methods is that, despite measuring protein properties in different contexts using various experimental protocols, they are measuring an inherent property of the proteome. We acknowledge that the raw values for stability measured across different contexts may (and should) vary. However, we believe the relative stability of a given protein within the human (or mouse) proteome should be comparable. Thus, we measured the correlation of the different datasets following normalisation and have now updated Figure 1 to show the distribution of these normalised values.

3. The TME_MI reagent was previously mainly used to detect the foldedness of the total proteome but less so to derive mechanistic insights in unfolding events with protein and sub-protein resolution as applied here. This comes with an increased need to provide evidence that the label does not lead to unfolding and localized structural rearrangements by itself, e.g. due to the excessive hydrophobicity of the modification compared to the free cys. Validation with different reagents or independent methods (such as proteolysis) would be needed to back the claims made for individual proteins (e.g. HSP8).

We acknowledge that the TPE-MI reaction may change the features of the proteins targeted. This is true of any approach to label proteins or measure a feature of them (including limited proteolysis, thermal denaturation and methionine oxidation). There is no reason to expect that TPE-MI would be more or less problematic than any of the other approaches. However, our data on β -lactoglobulin unfolding (Fig S1) establishes that TPE-MI can report accurately on its unfolding. Further support for our take home message is provided from the use of independent methods on HSPA8. Indeed, what we have done on HSPA8 was intended to bookend the take home message (rather than be the focus of the work). The key points of our study relevant to the point are that the TPE-MI data correlates with the previously published methionine oxidation data in unmasking insight to chaperone activity in urea denaturation curves of lysate. The new figures we have included showing the identification of other patterns of cys protection in other chaperones, and the effect of the HSP70 inhibitor on cys reactivity in the urea denaturation curve, now further corroborates that take home message.

4. Likewise, comparisons to methionine oxidation studies need to be taken with a pinch of salt as oxidation can have profound and long ranging structural impact that is difficult to distinguish from direct local unfolding in differential experiments (e.g. <https://www.sciencedirect.com/science/article/pii/S092544390900297X>)

As indicated above we acknowledge that any reactive approaches have the potential to induce structural change in distant regions of a protein. As described above, the strong correlation with the met oxidation to the TPE-MI reactivity provide confidence that we are

detecting events applicable to the stimulus rather than the TPE-MI treatment. We have revised the manuscript to acknowledge the potential for these effects to confound labelling studies in the discussion (pg 13).

5. At high concentrations of denaturants difficult to control parameters include the denaturing effect of covalent ligands such as the TPE-MI reagent and increasing oxidation effects that may make individual residues appear less reactive to other reactants. Hence, it is important to quantify the probe reacted and the non-probe reacted fraction of cysteine residues of interest to keep control of the total balance of the monitored cys residues. This is particularly important when trying to explain complex reactivity patterns such as the ones for HSP8 NBD. Unfortunately, the described approach only uses the changes in abundances of peptides with unreacted cysteins for quantification only (p4). - Is this because of the TPE-MI label causes a strong hydrophobic shift and unfavorable fragmentation? Please discuss.

This is correct, TPE-MI modified peptides are not readily monitored via mass spectrometry due to their unfavourable chromatographic properties, and thus we monitor loss of the unmodified cysteine-containing peptides. While we cannot distinguish TPE-MI from oxidative modification of individual peptides, both result in a net loss of the unreacted peptide thereby reporting on the sum change in the reactivity for that peptide. As noted by the reviewer, the total balance of the monitored cys residues is obtained at the protein level by normalization to non-cys peptides. In addition, the lysate for each reaction in a single biological replicate was extracted from a single mixed population of cells. Therefore, the starting balance of modified to unmodified cysteines should be identically distributed. A discussion of these details has been included on pg 4.

6. I have strong reservations with the interpretation of clusters in Figure 2B: Cluster 4 looks like cluster 1: unchanged reactivity with just a bit more variable data and potentially a slight increase around 1 M. Similarly, clusters 2 and three are essentially the same just that 3 contains more low quality data with a similar but opposite effect around 1 M urea. For 1 and 4 no wonder that the fits are rather poor as there are essentially no changes, for clusters 2 and 3 only goodness of fit distribution is somewhat different. For clusters 1 and 4 concordance of no change would be more informative than the correlation. The dataset shown in Figure 3E that was derived from a different publication seems to be of better data quality and Cluster 4 indeed indicates reduced reactivity of methionine residues. It should be noted though that methionine labeling is even more susceptible to oxidation artifacts than Cys labeling.

Our approach to the analysis was to be agnostic to the clustering. The clusters that resulted (and hence in Fig. 2B) were derived from unsupervised application of the fuzzy-c means algorithm; an approach for computational clustering in which a series of cluster numbers were evaluated using the elbow method to allow agnostic cluster partitioning. The small differences described by the reviewer in comparing pairs of clusters (i.e. "potentially a slight increase around 1 M", and "similar but opposite effect around 1 M urea") are precisely the subtle differences normally missed by brute-force application of denaturant curve fitting and is indeed a key feature of our study. While we acknowledge that the TPE-MI dataset is noisy, the appearance of similar cluster patterns in the independent methionine-oxidation dataset validates the distinction of these clusters within our dataset. The biological relevance of individual proteins' membership to a given cluster is open to interpretation. However, this regime has allowed us to distinguish between unfolding and protection that has been supported by the chaperone assays.

We have revised our descriptions of the clusters to better reflect how it was meant to be an agnostic analysis.

7. Figure 3 not particularly informative. D is kind of self-explaining as larger proteins tend to have more domains and can interact with more proteins than shorter ones. The entire figure might be better placed in supplemental.

We believe it is worth showing the quantitative evidence of this inference and also for following through with an agnostic analysis of the data, even if it seems at face value to be self-explaining. The figure also serves to compare between individual clusters as well as between unclustered and multiclustered proteins, an important contribution to our understanding of what molecular properties might predispose a peptide to a particular denaturation phenotype.

8. HSPA8 is a very complex protein to study: due to its roles in the cytosol but also in the lysosome as main carrier for chaperone mediated autophagy. Depending on experimental protocols access to the different subcellular compartments might be different at different denaturant conditions. This affects argumentation provided in line 253 ff for example. Changes observed for individual Cys residues are very small and even for the just significant ones the differences of the mean are much below 0.1. For Cys17 spread is just bigger so one cannot conclude that the data for 17 are significantly different to those for 574. Whilst the conclusions may seem plausible, they lack validation.

As per the point 1 from reviewer 2, we have performed a new experiment to probe the reactivity of cysteines in HSPA8 upon urea denaturation, which provides results from a different method that support the same key conclusion. We appreciate that the HSPA8 is a difficult protein to study and its complexities or heterogeneity of molecular functioning may indeed explain why we are detecting small changes on average. Nonetheless, we have applied conservative statistical models to make the points that are made, so that we do not over interpret what we have found. We stress that the investigation of the changes in HSPA8 are meant to highlight the main point of the paper, which is to illuminate how residue labelling approaches can be explained by functional responses to a stress. One of the exciting findings from our study is that it highlights the potential for a more detailed investigation into the structure and function of HSPA8 (and indeed the other chaperones we now highlight in the revised manuscript in Fig 6) in its natural setting. A more robust validation beyond what we have done and tried to do already with the crosslinking (Reviewer #2 point 2 above) would be well suited to such an investigation.

Minor point

1. Figure 4 could do with a better structural model of HSPA8. It might be worth including the one from Figure S6. Suggest depicting the various nucleotide binding motifs as well.

We have now moved the structural models from Fig S6 into the main figure as Fig 4A-C as suggested.

Reviewer #4 (Remarks to the Author):

This manuscript describes an approach to identify information on protein conformation and ligand interactions from denaturation data. The study determines that datasets for published denaturation studies do not only provide information on protein folding but also contain information regarding other structural aspects. Further, the authors utilized thiol reactivity labeling to further underscore that these types of studies provide more information than just on protein folding. In particular, this study demonstrated that cysteine peptide reactivity profiles could be defined by four distinct clusters, two of which closely resemble a two-state unfolding process. A decrease in cysteine reactivity in peptides in cluster 4 show other mechanisms of structural changes occurring such as interactions or conformational changes that are revealed in the data. The experimental design is well thought out and the proper controls have been analyzed. However, there are some significant changes in the manuscript that need to be made prior to publication.

1. The Methods sections states that 12 studies were deemed “suitable” for the pairwise analysis. What was the criteria used to make these 12 “suitable?”

Abstracts were filtered manually for those which (i) reported primary experimental data for proteome-wide protein stability under control conditions, (ii) were derived from either human or mouse samples, and (iii) provided a publicly-accessible summary of the dataset. We have clarified the language surrounding this in the methods (Supplementary Information pg 2).

2. On page 6 line 161, it states clusters 2 and 3 were likely to contain hydrophobic and solvent-buried residues. This is the case for cluster 2 in Figure 3B but not so for cluster 3 when it comes to buried residues.

This was an oversight, and have updated the text accordingly (pg 6).

3. The authors should provide some sentences to explain PFAM where it is first mentioned on page 6 line 169. This term may not be commonly known.

This was an oversight, and have updated the text accordingly (pg 6).

4. Page 7 line 186 refers to Figure S3E, I believe this should be Figure S2E

This is correct, we have updated the text accordingly (pg 7).

5. Page 8 line 213 refers to Figure 3G, I believe this should be 3H

This is correct, we have updated the text accordingly (pg 8).

6. Page 8 line 238 describes residues Cys267 and 269 as having overall decreased reactivity. It appears from figure 4A that the reactivity is actually increased. The authors need to explain this.

This was an error, and we have updated the text to reflect that the peptide was in fact increased (pg 9).

REVIEWERS' COMMENTS

Reviewer #1 (Remarks to the Author):

The revised manuscript has adequately addressed the concerns of this reviewer.

Reviewer #2 (Remarks to the Author):

The authors have addressed all of my concerns in the revised manuscript (and in the response to reviewers). The manuscript is now much improved and I recommend it for publication in Nature Communications.

Reviewer #3 (Remarks to the Author):

In the revised manuscript, `Hidden information on protein function in censuses of proteome foldedness` the authors have addressed many of my previous concerns, added more data, and importantly now provide a more balanced discussion of their data and published findings.

This reviewer supports publication of the amended manuscript.

Minor:

Line 25 typo, should be `outlined`.

Line 136 sentence needs revision

Reviewer #4 (Remarks to the Author):

The authors have successfully responded to my concerns. I recommend the article for publication.